

**Topography- and nightlight-based national flood risk assessment in Canada**
Amin Elshorbagy[1,2], Anchit Lakhanpal[1], Bharath Raja[1], Serena Ceola[3], Alberto Montanari[3],
Karl-Erich Lindenschmidt[2,4]
[1]Department of Civil, Geological, and Environmental Engineering, University of Saskatchewan, Saskatoon, Canada.
[2]Global Institute for Water Security, University of Saskatchewan, Saskatoon, Canada.
[3]Department of Civil, Chemical, Environmental, and Materials Engineering, University of Bologna, Bologna, Italy.
[4]*School of Environment and Sustainability, University of Saskatchewan, Saskatoon, Canada.*
*Correspondence to: Amin Elshorbagy (amin.elshorbagy@usask.ca)*
**Abstract**
In Canada, flood analysis and water resource management, in general, are tasks conducted at the
provincial level; therefore, unified national-scale approaches to water-related problems are
uncommon. In this study, a national-scale flood risk assessment approach is proposed and
developed. The study focuses on using global and national datasets available at reasonably fine
resolutions to create flood risk maps. First, a flood hazard map of Canada is developed using
topography-based parameters derived from digital elevation models namely Elevation Above
Nearest Drainage (EAND) and Distance From Nearest Drainage (DFND). This flood hazard
mapping method is tested on a smaller area around the city of Calgary, Alberta, against a flood
inundation map produced by the City using hydraulic modeling. Second, a flood exposure map of
Canada is developed using a land-use map and the satellite-based nightlight luminosity data as two
exposure parameters. Third, an economic flood risk map is produced, and subsequently overlaid
with population density information to produce a socioeconomic flood risk map for Canada. All
three maps of hazard, exposure, and risk are classified into five classes, ranging from very low to
severe. A simple way to include flood protection measures in hazard estimation is also
demonstrated using the example of the city of Winnipeg, Manitoba. This could be done for the
entire country if information on flood protection across Canada were available. The evaluation of
the flood hazard map shows that the topography-based method adopted in this study is both
practical and reliable for large-scale analysis. Sensitivity analysis regarding the resolution of the



digital elevation model is needed to identify the resolution that is fine enough for reliable hazard
mapping, but coarse enough for computational tractability. The nightlight data are found to be
useful for exposure and risk mapping in Canada; however, uncertainty analysis should be
conducted to investigate the effect of the overglow phenomenon on flood risk mapping.
**Keywords:** flood hazard, exposure, risk, nightlights, Canada.
**1 Introduction**
Rivers, and water bodies in general, have always been the most attractive landscape feature for
humankind. Historically and to date, rivers have provided people with water for drinking and
agriculture, food, an inexpensive mode of transportation, a natural drain for their effluents, and
fertile land for agriculture in the floodplains. Consequently, most populous cities in the world are
built around rivers. Interestingly, even recent studies show that people are still moving closer to
streams in various regions of the world (Ceola et al. 2015). The increased flood hazard comes as a
natural consequence of encroaching on floodplains.
Globally, floods are among the most feared natural hazards as they can inflict large scale economic
and social damage, cause panic, and disrupt essential services. Annually, thousands of lives are
lost due to floods, with 5200 lives, for example, claimed in 2011 alone (Balica et al. 2013). The
most recent 2016 floods in Louisiana, USA, claimed 13 lives and left 40,000 homes under water.
In Canada, flood damages exceeded 7.4 billion US dollars over the recent five years (2010-2015),
with 9 lives lost and more than 100,000 individuals directly affected, according to the
CRED/OFDA International Disaster Database (http://www.emdat.be/database). This has led the
Canadian government to establish FloodNet – a Canada-wide strategic research network for flood
forecasting and impact assessment.



Floodplains and low-lying lands are typically areas with high levels of flood hazard due to their
elevation and proximity to rivers; however, society makes such areas more exposed by inhabiting
them and establishing valuable economic investments, with insufficient measures to contain
vulnerability in most cases, and thus, increasing flood risk as the product of hazard, exposure, and
vulnerability (Balica et al. 2013; UNISDR, 2009; Samuels et al., 2009). Some argue that areas that
have not been flooded for a long time tend to be encroached by the society, causing the damage
from future floods to be higher than expected, whereas areas that were recently damaged by floods
seem to encounter lower than expected damages when another flood occurs (Di Baldassarre et al.
2015). It has been suggested that social memory plays a significant role in flood vulnerability as
societal preparedness can be different based on the recent history of floods. This emphasizes the
importance of developing a systematic flood risk assessment approach that helps societies,
insurance companies, water managers, and policy makers make informed decisions.
National flood risk assessment approaches are useful but challenging as data required to develop
realistic approaches can be extensive, and detailed hydraulic modeling without proper
prioritization of high risk areas can be unjustifiably costly. In recent years, there has been an
increasing use of remotely sensed and global datasets in water resources as they can make such
studies on a national scale possible. For example, GRACE (the Gravity Recovery and Climate
Experiment) has been shown to provide data on water cycle and groundwater reserve that are
needed for water management (Famiglietti and Rodell, 2013). Satellite-based data, e.g., snow
cover data, have proven valuable for calibrating hydrological models (Parajka and Bloschl, 2008)
and for flood detection and mapping (Brakenridge and Anderson, 2006). Ceola et al. (2014; 2015)
used 1-km resolution nightlight datasets to show human interaction with streams as well as
exposure to floods, based on the fact that nightlights reflect human activities. As nightlights can





indicate the spatial distribution and temporal trends, in certain regions, of human activities around
rivers, we reiterate that they are of obvious relevance to flood risk assessment studies, especially
on a large scale.
Ceola et al. (2014) relied mainly on the proximity of population to rivers to assess exposure to
floods. However, a research question that has been left unaddressed by previous studies that used
nightlights relates to the datasets that are needed, in combination with nightlights, to establish flood
risk assessment approaches that are realistic and feasible. The aim of this study is to integrate
several and relevant sources of information to develop a flood risk assessment approach for
Canada, which will lead to national flood hazard and risk maps that benefit from topographic
information, remotely sensed nightlight data and, as an option, local information to estimate
vulnerability. The end product should be flexible, easily updatable, and help stakeholders assess
areas that require further attention through, for example, detailed hydraulic modeling.
**2 Flood hazard, exposure, vulnerability, and risk**
The terms of flood hazard, exposure, vulnerability, and risk are sometimes confusing to readers as
they may have different meanings for different users. The four terms may even be used
interchangeably to refer to the same thing. Following the definition provided by UNISDR, 2009;
IPCC, 2012; Colleantuer et al., 2015, flood risk is given by a combination (e.g. the product) of
hazard, exposure, and vulnerability (Equation 1).
Flood risk = flood hazard $\times$ flood exposure x flood vulnerability        (1)
*Hazard* is used by some researchers to mean the flood disaster itself or its potential occurrence
(Gilard, 2016; UNISDR, 2009, Colleantuer et al., 2015), identified more precisely (Sayers et al.,
2002) by two main components – source (e.g. rain) and pathway (e.g. flood extent and depth). This



definition is appropriate and usually quantified from an engineering perspective as the probability
of occurrence of a flood event (Balica et al., 2013; de Moel et al., 2009). Intuitively, a low-lying
area that is close to a river has a higher level of flood hazard (impacted by more frequent floods)
than an area of higher elevation that is far-removed from the river. In this study, *distance* from,
and *elevation* above, the river are used as two indicators of the flood hazard level of any land pixel.
Exposure (i.e. elements at risk) is given by the economic and intrinsic values that are present at
the location involved (IPCC, 2012). Population density, capital investment, and land or property
value can be indicators of flood exposure. *Vulnerability,* following Adger (2006) and Colleantuer
et al. (2015), is defined as the capacity of the society to deal with the flood event, namely the state
of susceptibility to harm from exposure to an undesired event, floods in this study, associated with
environmental and social change and lack of capacity to adapt. Lack of flood defenses or protection
of economic values and human lives susceptible to floods are indicators of vulnerability.
Obviously, the product of exposure and vulnerability reflects an integrated measure of the
environmental and socioeconomic consequences of floods. The main reason for the increase in
losses due to floods is the increase in the population and people's preference to reside in flood
prone areas, which makes them exposed to floods (Jonkman, 2005; Ceola et al. 2014). An example
of the policy and social dimension of exposure is depicted in Figure 1 for the city of Fort
McMurray, Alberta, Canada, which shows how the society encroached into areas of higher level
of flood hazard over the years. The increase in exposure is indicated by the spatial expansion and
increase in nightlight luminosity from 1999-2013, which is considered a proxy for socioeconomic
activities (Doll et al. 2000). For the simplicity of display, the flood hazard map was classified into
three levels of hazard based on elevation above and distance from the nearest rivers. Land-use,
nightlight, and population are used in this study as indicators of flood exposure.



In the literature, frameworks or guidelines for flood risk assessment at the national level are
limited. A classic example is the work of Hall et al. (2005), who conducted a national-scale flood
risk assessment in England and Wales for the purpose of prioritization of resources for flood
management. The methodology of Hall et al. (2005) benefited from rich information available on
the standard of protection, condition and location of flood defences, as well as flood extent maps,
occupancy, and asset values in England and Wales. de Moel et al. (2009) noted that flood extent
maps are the most commonly produced flood maps in Europe, and that only very few countries
have developed flood risk maps that comply with the European Directive (2007/60/EC). Later,
Lugeri et al. (2010) developed a flood hazard map of Europe, identifying low-lying areas adjacent
to rivers, and used it with land-use data and a damage-stage relationship to identify flood risk. A
coarse global scale flood risk assessment was also developed by Ward et al. (2013) using global
hydrological and hydraulic modeling. The work presented in this paper is at a finer resolution, and
using different types of data based on topography and remotely sensed, which lead to a low-cost
flood mapping product that is relevant at a national scale.
The level of detail required for flood risk analysis is an important issue, which is obviously related
to the spatial scale of the study area. Even in urban areas, Apel et al. (2009) found that a medium-
level complexity model for both hazard and exposure is sufficient. One could expect that on
national scale for large countries, aggregate measures and index-based approaches might be the
feasible choice. When compared with a physically based modeling approach, a parametric
approach, which uses flood hazard and exposure indices, can direct decision makers to simplified
usage and simpler understanding of the risk, and thus, better allocation of resources and
investments for flood management and protection (Balica et al. 2013).





As the second largest country in the world, the continental extent of Canada from 41.7º to 83.111ºN
and from 52.619º to 141.010ºW, encompasses different topographies ranging from flat prairies to
mountains and different climates from semi-arid to wet. On an average annual basis, Canadian
rivers discharge 9% of the world's renewable water resources (Whitfield and Cannon, 2000).
Fluvial floods in Canada can happen as a result of excessive rainfall, similar to the 2013 flood in
Alberta, however, high water levels often result from reduced channel capacity due to ice and
debris jams (NRCC, 1989). Therefore, water levels and extent of floods may not reflect the
conventional return period associated with the flood discharge. Floods are usually monitored,
analyzed, and managed at the provincial level, which makes a Canada-wide unified flood
modeling, mapping, and analysis, as well as flood-related data accessibility laborious tasks.
**3 material and methods**
To develop a national-scale framework for flood risk assessment in Canada, parameters
representing the concepts of hazard and exposure were identified and subsequently, a flood risk
index was developed based on the integration of both hazard and exposure. All three types of maps
– hazard, exposure, and risk – are presented separately as they each contain distinct and useful
information. In a subsequent step that is developed for the city of Winnipeg, Manitoba, we show
how flood protection measures, as might be represented within hazard or vulnerability, can be
incorporated.
**3.1 Hazard parameters and mapping**
It is common to define and classify flood hazard based on flood magnitude and/or frequency (Apel
et al. 2009; Balica et al. 2013), but classification based on depth is also used (Masood and
Takeuchi, 2012). The frequency and magnitude of floods, along with their associated inundation



depth, are constantly changing due to economic development and climate change (Milly et al.
2002), which challenges the estimates and definition of flood hazard and risk on a range of scales
(Merz et al. 2010a). Therefore, classifying hazard levels on a national scale based on topography
(Lugeri et al. 2010) is both realistic and sound, as it can be converted locally to other types of
classification as will be discussed here in the results section.
In this study, flood hazard was estimated using two parameters: elevation above the nearest
drainage (EAND), which is similar to HAND (height above nearest drainage, Rennó et al. 2008)
and distance from the nearest drainage (DFND). These two parameters define the topography of
an area and thus, help in determining the relative position of a place with respect to the stream.
Both parameters were derived from a Canadian digital elevation model (DEM) obtained from
Natural Resources Canada (http://geogratis.gc.ca/site/eng/extraction). A coarse 326 m DEM was
used for the Canada-wide analysis in this study to keep the computational cost manageable;
however, a comparison was made with a much finer resolution DEM (20 m) for an area of the city
of Calgary, Alberta, to evaluate the effect of the DEM's resolution on the flood hazard map.
EAND is a terrain descriptor, which produces a new normalized DEM where pixel values represent
altitudes relative to the local drainage instead of the mean sea level. To allocate elevation values
to the pixels with respect to local drainage, we first identified the drainage network by using the
ArcGIS hydrology tool. The DEM, available in raster format, was initially filled by identifying
pits and raising their elevation to the level of the lowest pour point. After obtaining the filled DEM,
the second step was to generate flow direction. There are a total of eight valid output flow
directions, corresponding to the eight adjacent cells into which water may flow. The flow direction
tool follows eight directions flow model, which was presented by Jenson and Domingue (1988).
After identifying the drainage network for Canada, a new raster was created using the Euclidean



allocation tool available in the spatial analyst toolbox of ArcGIS. All pixels within this raster were
assigned the new values of elevation, which were the elevation values of the nearest drainage pixel
based on Euclidean distance. Finally, this output was subtracted from the original elevations to
obtain the EAND map for the study area. Also, for each pixel, the DFND – the horizontal distance
from the nearest drainage network – was calculated. Negative values of EAND could be observed
because there were depressions lower than the nearest stream. EAND and DFND were classified
into five different EFND and DFND classes as shown in Table 1. The lower values of EAND and
DFND were assigned the higher class values as they indicate the low-lying and close areas to the
streams, respectively, and thus, the highest level of flood hazard. The hazard value was calculated
based on the product of EAND and DFND classes; e.g. a hazard level of 20 could result from
EAND class 4 and DFND class 5 (or vice versa). Finally, hazard values were reclassified into five
different hazard classes as shown in Table 1. The class intervals were selected somewhat arbitrarily
in this study. However, depending on the topography of the study area, other hazard class intervals
can be selected.
**Table 1. Classes of elevation above nearest drainage (EAND), distance from nearest drainage**
**(DFND), and the resultant flood hazard for Canada.**

| EAND (m) | Class | DFND (m) | Class | Hazard | Class | Hazard level |
|----------|-------|----------|-------|--------|-------|--------------|
| ≤ 2.0 | 5 | ≤ 1000 | 5 | 21 – 25 | 5 | Severe |
| 2.1 – 4 | 4 | 1001 – 2500 | 4 | 16 – 20 | 4 | High |
| 4.1 – 6 | 3 | 2501 – 5000 | 3 | 11 – 15 | 3 | Medium |
| 6.1 – 8 | 2 | 5001 – 10000 | 2 | 6 – 10 | 2 | Low |
| > 8.0 | 1 | > 10000 | 1 | 1 – 5 | 1 | Very low |

The topography-based hazard mapping approach developed in this study was evaluated against
flood inundation map developed using hydraulic modeling by the city of Calgary (Government of
Alberta, 2013) for an area of Calgary to evaluate the utility of our approach. Another important
parameter that affects the flood and its impact on the floodplain is the existence of flood protection



or defence measures. Including flood protection within hazard or vulnerability can be debatable.
However, the approach we adopt in this study depends on the type of the flood protection.
Structural flood protection measures that affect the flood runoff itself (Mays, 2011), such as dikes
and dams, are included within hazard assessment as they affect the flood stage-discharge and
discharge-frequency relationships. Non-structural measures, such as zoning, insurance,
rearranging spaces, and raising buildings, are included within vulnerability assessment because
they affect the susceptibility of the floodplain (UNISDR, 2009) rather than the flood water (Mays,
2011). When such information on flood protection is available for the whole country, flood
protection can be included as the third hazard parameter to identify the final hazard level or as a
separate vulnerability parameter. Flood protection can be included as a binary parameter, i.e.
protected/unprotected or in the form of various levels of protection. For the current study, complete
information on flood protection across Canada was not made available to us; however, we
investigated how to consider protection on a smaller regional scale around the city of Winnipeg,
Manitoba, and it will be shown in the results section.
**3.2 Exposure parameters and mapping**
As reflected in most flood studies, there is no doubt that land-use is the most relevant flood
exposure parameter as it indicates the land or property value, e.g. urban development or
agricultural land. In this study we also used a *land-use* map for Canada available through the North
American Land Change Monitoring System (NALCMS; Latifovic et al. 2012), which is available
in raster format at a spatial resolution of 250 m and can be obtained through
http://www.cec.org/tools-and-resources/map-files/land-cover-2005. The original land-use data
taken from NALCMS define 19 land-use types for North America, out of which there are 15 types
found in Canada. These types were further reclassified for the purpose of this study into five types





as shown in Table 2. There are no agreed upon global rules for land-use classification, however,
for the purpose of national-scale flood risk assessment, these five types were judged to be
sufficient, and also bear some similarity to the European Corine Land Cover classes
(http://uls.eionet.europa.eu/CLC2006/CLC_Legeng.pdf). The reclassified land-use types were
then assigned values between 1 and 5 according to their economic value, with the values of 5 and
1 assigned to urban areas and water bodies, respectively.
The second flood exposure parameter considered in this study is *nightlights*. Nightlight satellite
imagery has been investigated as a proxy for human activities, and has been used in various studies
from different domains (Raupach et al., 2010; Zhou et al., 2014; Gomez et al., 2015; Townsend
and Bruce, 2010). Ceola et al. (2014) explored nightlights to examine human exposures to floods
worldwide, using HydroSHEDS data, based only on proximity to streams. The study included 175
regions covering 168 countries with the exception of Canada, Russia, and part of northern Europe.
The nightlight values, defined by a digital number (DN) ranging from 0 to 63 to reflect the degree
of luminosity, were classified for Canada into five different nightlight classes (NC) as shown in
Table 3. The nightlight data were obtained from the National Oceanic and Atmospheric
Administration (NOAA) of the United States
(http://ngdc.noaa.gov/eog/dmsp/downloadV4composites.html). The spatial resolution of the
dataset is 30 arc-seconds (corresponds to roughly 1 km near the equator) and the data are available
for the period 1992-2013. The most recent available data of 2013 were used for our analysis, and
the Canadian nightlight map of the year 2013 is shown in Figure 2. Usually data from more than
one satellite are available and, similar to Ceola et al. (2014; 2015), the average values of all
satellites were used in this study.





**Table 2. Classes of land-use types in Canada along with their percent of area covered.**

| Land-use type | Reclassified land-use | Land-use class (LC) | % of area covered |
|---|---|---|---|
| - Wetland *(marshes, swamps, mangroves);*<br>- Water *(open water);*<br>- Snow and Ice *(perennial cover)* | Water bodies | 1 | 16 |
| - Barren land;<br>- Sub polar or polar barren moss<br>- Temperate or sub-polar grassland;<br>- Sub polar or polar grassland | Wasteland/ Grassland | 2 | 28.2 |
| - Temperate or subpolar needle leaf forest;<br>- Temperate or subpolar broad leaf forest;<br>- Mixed forest;<br>- Temperate or subpolar shrub land;<br>- Subpolar or polar shrub land | Forest | 3 | 50 |
| Cropland | Agriculture | 4 | 5.7 |
| Urban and built up | Urban | 5 | 0.1 |

**Table 3. Classes of nightlight luminosity in Canada from 1 – 5. The exposure classes were**
**selected based on the product of nightlight and land-use classes.**

| Nightlight value (DN) | Nightlight class (NC) | Nightlight level | % area covered | Exposure | Class | Exposure level |
|---|---|---|---|---|---|---|
| 0 – 5 | 1 | Very low luminosity | 93.6 | 1 – 5 | 1 | Very low |
| 6 – 10 | 2 | Low luminosity | 4.4 | 6 – 10 | 2 | Low |
| 11 – 30 | 3 | Medium luminosity | 1.4 | 11 – 15 | 3 | Medium |
| 31 – 59 | 4 | High luminosity | 0.5 | 16 – 20 | 4 | High |
| 60 – 63 | 5 | Very high luminosity | 0.1 | 21 – 25 | 5 | Severe |

The ranges of the first two classes (having DN ≤ 10) were kept narrow because they are spread
over most part of Canada (about 98% of Canada's area). They indicate absent or low human
activity and, hence, from a flood exposure perspective they are less important. Accordingly, low
nightlight class values were assigned to them. The range of DN values 11-30 is significant as it is
mainly found in parts of the forest and agricultural land that possess more important resources than





the first two classes. The DN range of 31-59 is found in the outskirts of cities and towns, and
represents mostly agricultural lands and small establishments. The pixels having DN values of 60
and above fall within city boundaries and contribute up to 80% of the nightlights of the city.
Therefore, 60 and above were kept as a separate class (NC=5), highlighting urban centers, which
are the most flood exposed areas. Similar to the calculation of the hazard index, exposure was also
calculated as the product of land-use and nightlight classes, leading to values ranging from $1 - 25$.
The exposure values were further reclassified into five classes as shown in the last three columns
of Table 3, and a flood exposure map of Canada was produced.
Finally, and based on equation (1), flood risk was calculated as the product of hazard and exposure,
as local vulnerability information was not available, and was reclassified into five risk classes as
shown in Table 4. In the absence of population data, nightlights might be taken as a surrogate for
population. However, our investigation reveals that both datasets may differ in some places. This
is expected as nightlights are more representative of economic investment and activities, which
can be different from population. For example, airports and industrial and commercial areas are
highly luminous but the census data show low or no population. Moreover, population data,
especially when associated with qualifiers regarding different groups and income can be
distinctively used to assess social vulnerability or exposure to floods (Adger, 1999). As floods may
have different impacts on the relative well-being of individuals and groups, which is not reflected
by classic economic exposure, it is important to identify the impact of floods on population
separately, without integrating or averaging with other exposure parameters. Therefore, in this
study the physical flood risk map of Canada was produced first, then it was overlaid with the
population information to allow reclassification of the risk map based on the distribution of
population.





**Table 4. Classes of flood risk in Canada, which results from the product of hazard and**
**exposure.**

| Flood risk value | Risk class (RC) | Risk level (RL) |
|---|---|---|
| 1 – 5 | 1 | Very low |
| 6 – 10 | 2 | Low |
| 11 – 15 | 3 | Medium |
| 16 – 20 | 4 | High |
| 21 – 25 | 5 | Severe |

**4 Results**
**4.1 flood hazard mapping**
The topography-based (EAND and DFND) flood hazard map of Canada, developed and classified
based on the method explained in the previous section, is shown in Figure 3. Large areas of the
country are classified under high and severe levels of flood hazard due to their low elevation and
proximity to rivers. However, most of these areas have negligible human presence and economic
investments. The flood hazard map can be useful for large-scale planning and development, where
avoiding encroachment into flood hazardous area is recommended. In support of identifying the
flood information needed for flood insurers to assess their exposure to floods and to price the flood
elements at risk, Sanders et al. (2005) identified the availability of fine-resolution DEMs as the
key obstacle for such analysis. For the national-scale analysis in this study, we used the 326 m
DEM resolution, as it is computationally tractable for a country like Canada. However, a
comparison between hazard mapping using the 326 m and a much finer resolution of 20 m was
conducted on a smaller area around the city of Calgary, Alberta. Even though Figure 4 shows an
overall reasonable visual match between both flood hazard maps produced using the different
resolutions, there are important differences to be observed. The stream network itself, generated
using the DEMs, can have significant differences, depending on the resolution. Figure 4(b),
produced using the 20 m DEM, shows a more realistic representation of the Elbow River and its



confluence with the Bow River, compared to ground truth, and thus, a more reliable flood hazard
assessment for the area around downtown Calgary. Other areas, such as the top right corner of the
city, appears only with the 326 m resolution as an artefact of the coarse DEM's resolution.
Depending on the purpose and use of the flood hazard map, caution must be exercised with regard
to the adopted DEM.
A flood inundation map of an area in the city of Calgary was produced by the City (Government
of Alberta, 2013), based on a 100-year flood determined by flood frequency analysis and using the
hydraulic model HEC-RAS. A comparison between the topography-based flood hazard mapping
method adopted in this study and the hydraulic modeling-based 100-year inundation map is shown
in Figure 5. There is good agreement between the model-based 100-year flood inundation (shown
as hatched grey area) and the hazard level classified in this study as *severe* (Table 1). Two sections
of the Bow and Elbow Rivers are enlarged, as examples, for better visual comparison between
both methods. As shown in the main map (on top) in Figure 5, there is good agreement in other
sections as well, and there are small areas that do not match well. Some smaller areas of the 100-
year flood are extended over the second highest hazard area defined in this study as *high*. This was
expected, as our classes shown in Table 1 were selected somewhat arbitrarily across Canada. The
hazard levels can be reclassified locally based on different values of EAND and DFND to match
particular floods, e.g. 100-year, 200-year, in areas where flood inundation using hydraulic
modeling is available. This way, the flood hazard map can be converted into approximate flood
inundation maps for floods with particular return periods.
Another important flood hazard parameter, which was not fully implemented here due to lack of
information, is flood protection measures. However, an example using an area near the city of
Winnipeg, Manitoba, is shown in Figure 6. The city of Winnipeg is protected from Red River





floods using a floodway (appears in the figure in pink color) that carries part of the flood runoff
around the city, and a dike (appears in the figure in yellow color) that prevents flood surface runoff
from entering the city from the west side. The effect of flood protection of these structural
measures is handled in our flood hazard mapping method by identifying the flood depth up to
which the city is protected (flood design level), then assigning the design level to the DEM cells
in the protected area. A hazard map with and without flood protection for the city of Winnipeg is
provided in Figure 6, which shows the reduced level of flood hazard within the city limits. Usually,
there are backwater and other hydraulic effects on areas upstream of flood protection, and such
effects cannot be easily captured by the topography-based hazard mapping adopted here. Hydraulic
modeling is recommended to investigate the effects of flood protection measures on upstream
unprotected areas.
**4.2 Flood Exposure and Risk Mapping**
The flood exposure map of Canada, which integrates land-use and nightlight information, is shown
as Figure 7. The areas of higher exposure is mainly concentrated around major urban centers in
Canada. As expected, the exposure map is quite similar to the nightlight map (Figure 2b), because
the distribution of nightlight matches to a great extent the land-use map; for example, urban areas
are much more luminous than forests. However, it is useful to include both types of information
as some major capital investments, reflected by high luminosity, can be situated within larger areas
classified as agricultural, or forested areas. Also, some large parks with lower luminosity can be
found within the limits of urban areas. Furthermore, nightlights are quantified using the DN, which
helps in using them as a proxy for economic investment/damage calculations in the absence of
monetary values. It is important to note that one of the shortcomings of using nightlights is the
phenomenon of "overglow" (Doll, 2008) – areas of low luminosity shown with false high





luminosity due to reflections from surrounding areas with much higher luminosity. Small et al.
(2005) listed three major causes for this phenomenon: coarse spatial resolution, large overlap
between pixels, and errors in the geolocation.
By assuming that flood vulnerability is homogeneous over Canada, a flood risk map of Canada,
which results from the product of flood hazard and exposure only, is shown in Figure 8. Even
though severe and high flood hazard areas are spread spatially over the entire country, severe and
high flood risk areas are concentrated in urban centers in the southern part of Canada. Severe and
high flood hazard in northern areas assume lower levels of risk when integrated with lower levels
of exposure in the north due to lack of human activities and urban centers.
A key flood exposure and risk parameter, which was deliberately left out of the risk map, is
population. Using the example of the Greater Toronto Area in Ontario, Figure 9 shows the
differences that are represented by nightlights, population distribution, and land-use maps. The
airport area, indicated by a grey triangle, and an industrial area indicated by a grey circle, are
typical examples of urban/built-up areas (Figure 9c) with high economic investments that are
highly luminous areas (Figure 9a), but very low – close to zero – population (Figure 9b). This
confirms that nightlights and population distribution can differ at times, and it is important to
include both parameters, but without integrating them in order to avoid the "average" effect.
To identify flood impact on people (social impact) and separate it from economic impact, we
propose overlaying the flood risk map (Figure 8) with a population density layer. Figure 10 shows
an example of such reclassification of the flood risk map with and without population on a smaller
area (the city of Calgary, Alberta) for better visualization of the concept. The central part of the
city with high-rise buildings and high population density remains within the highest levels of flood
risk where both economic and social risks are at their highest levels. The northern and southern





parts, which are mainly commercial areas with lower population density and, thus, lower social
risk, assume reduced levels of overall flood risk (Figure 10b) in spite of having severe economic
flood risk (Figure 10a).
**5 Discussion**
Even though flood hazard, exposure, and vulnerability maps are all important, the flood hazard
map is of special interest to both the public and planners or decision makers. The flood hazard map
allows the public to assess the situation of their properties with respect to floods, whether the
property is residence, agricultural land, or commercial business. For planners and decision makers,
flood hazard maps allow for assessing areas of future development, or locations of strategic
establishments. As mentioned earlier, the flood hazard map developed in this study can be
reclassified or converted to inundation maps of floods with specific return periods, e.g. 100-year
flood, using hydraulic modeling, or even linked to particular recorded flood events, such as the
known 1979, 1997, and 2011 floods in Manitoba. In some areas, like the city of Calgary (Figure
5), 100-year flood extent almost matches our severe flood hazard class (< 2 m). In other regions,
and depending on the topography, the 100-year flood might cover two or three of the flood hazard
classes.
For prioritizing resource allocation and intervention for flood damage mitigation, flood risk is the
important indicator as it integrates hazard, exposure, and vulnerability, and reflects the spatial
distribution of expected damage. The general flood risk map, similar to Figures 8 and 10a, can be
used for prioritizing intervention and estimating compensations based on economic flood risk, but
flood risk maps with population, similar to Figure 10b, add an important sociopolitical dimension
because they indicate where certain levels of risk affect more or less people. This type of



socioeconomic flood risk map can be made public to collect feedback from all stakeholders.
Certain groups falling under reduced levels of risk may raise issues of particular social exposure
or vulnerability, and help water managers revise the classification or use differential spatial
weights to produce more realistic socioeconomic flood risk maps. This approach of engaging both
the public and water professionals in co-production of flood-related knowledge can be initiated
using the risk maps (Lane et al. 2011).
The simple way presented in this paper for considering the effect of flood protection on the hazard
(or vulnerability), and thus the risk, classification can be useful for quantifying the change in the
spatial distribution of flood risk. This might prove useful for comparing flood risk with different
types of societal risk, e.g. forest fires. This method allows for quick assessment of the value of
flood protection measures, and the locations of critical need for such measures.
It is important to note that there are various uncertainties associated with the nightlight and
topography-based approach suggested in this paper for flood risk assessment in Canada. The
DEM's resolution is an important criterion, and sensitivity analysis might be needed to identify a
resolution that is coarse enough for tractable computations, but fine enough for reliable
identification of the stream network and the various hazard classes. The available nightlight data
are of much coarser resolution (1 km) than the required DEM's resolution. This difference, along
with the uncertainty stemming from the overglow phenomenon, can cast some doubts on the
nightlight classification. Therefore, exposure and risk maps should be treated with caution when
analyzing small areas. Finally, it is relevant to note the importance of local information for the
estimation of flood hazard and vulnerability. While the flood risk map based on hazard and
exposure may provide important indications to identify critical areas, information on existing flood
protection is necessary in order to provide useful guidelines to decision makers. Therefore,





obtaining local information is a fundamental step that can be carried out only by effectively
cooperating with actors who have a refined knowledge at the local level, like for instance local
water managers.
**6 Conclusions**
The topography- and nightlight-based approach adopted in this study for flood risk assessment on
a national scale is both useful and practical. Without detailed hydraulic modeling, the flood hazard
map of Canada can provide a reliable preliminary assessment of the flood hazard level anywhere
in the country. This low-cost product can be used for early stages of development planning.
Identifying the flood hazard level of even areas such as wastelands might prove useful for planning
and management of activities like mining in remote and undeveloped areas. The flood risk map,
which integrates both hazard and exposure, including nightlights, is the most useful product as it
allows for evaluating the spatial distribution of the expected flood damage, and thus, can help in
prioritizing government intervention and strategic resource allocation. The risk map, which
typically reflects economic risk can be combined with population distribution maps to explicitly
identify the social risk dimension as well as overall socioeconomic flood risk. It was shown in this
study that nightlight luminosity and population distribution can differ at certain locations, and it is
beneficial to use both types of information for flood risk assessment.
The severe and high flood hazard areas in Canada are spread over all regions of the country;
however, the severe and high flood exposure and risk are concentrated in the southern part of the
country around urban centers. Complete information on flood protection across Canada should be
collected and integrated with the developed hazard and risk maps produced in this study in order
for these products to be considered complete and ready to use. Some sensitivity analysis regarding



the required DEM's resolution is needed to identify the resolution that is fine enough for reliable
hazard mapping, but coarse enough for computational tractability. Both DEM's resolution and the
nightlight's overglow phenomenon are possible sources of uncertainty in the maps produced in
this study. Attempts should be made in the future to quantify such levels of uncertainty.

**ACKNOWLEDGEMENT**

The financial support of NSERC through the strategic research network – FloodNet – and the
Discovery Grant program is acknowledged. SC and AM gratefully acknowledge the financial
support from the FP7 EU funded project SWITCH-ON (grant agreement n. 603587).

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

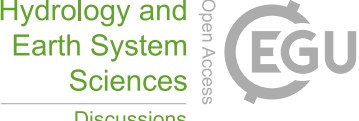


(a)

(b)

Figure 1: (a) Variation in nightlights around Fort McMurray, Alberta, Canada between the years
1992 – 2013 ($DN_{2013}$- $DN_{1992}$). Positive values indicate increase in nightlight luminosity; (b)
Classified flood hazard map for the same area. The expansion of human activities over high
flood hazard area are obvious.





(a)

(b)

Figure 2: Nightlights over Canada shown as (a) Continuous spectrum and (b) classified as shown
in Table 3 into very low luminosity (0-5), low luminosity (6-10), medium luminosity (11-30), high
luminosity (31-59), and very high luminosity (60-63).



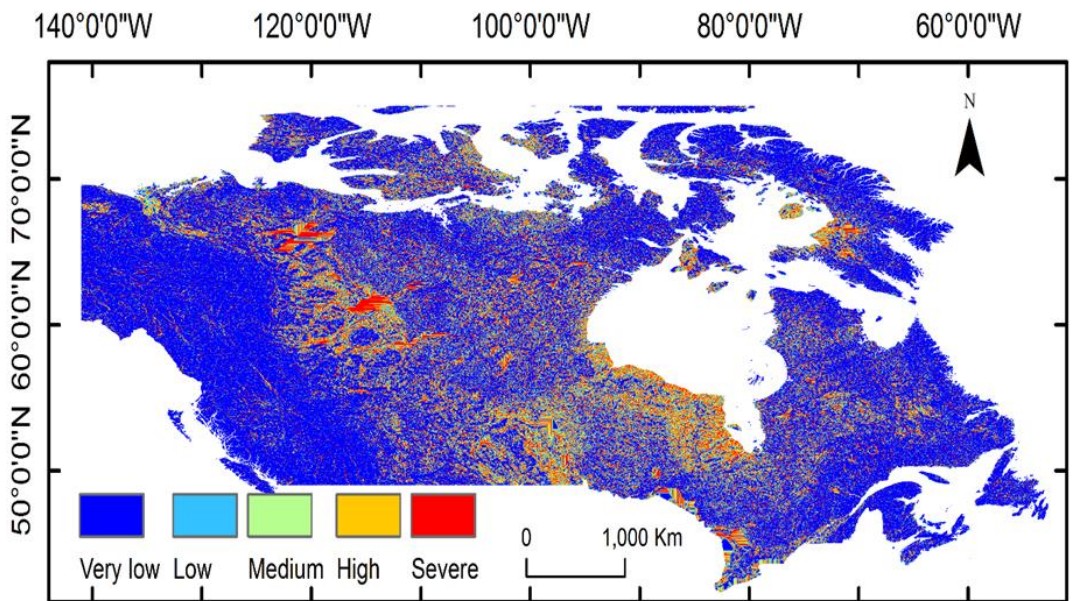

Figure 3: Flood hazard map for Canada obtained using the 326m DEM. Large areas are classified under high- and sever-level flood hazard, but most areas have negligible human presence and investments.





2    Figure 4: Effect of the DEM resolution on flood hazard map. Example of the flood hazard map
3        for city of Calgary using two different DEMs with a resolution of (a) 326m and (b) 20m.



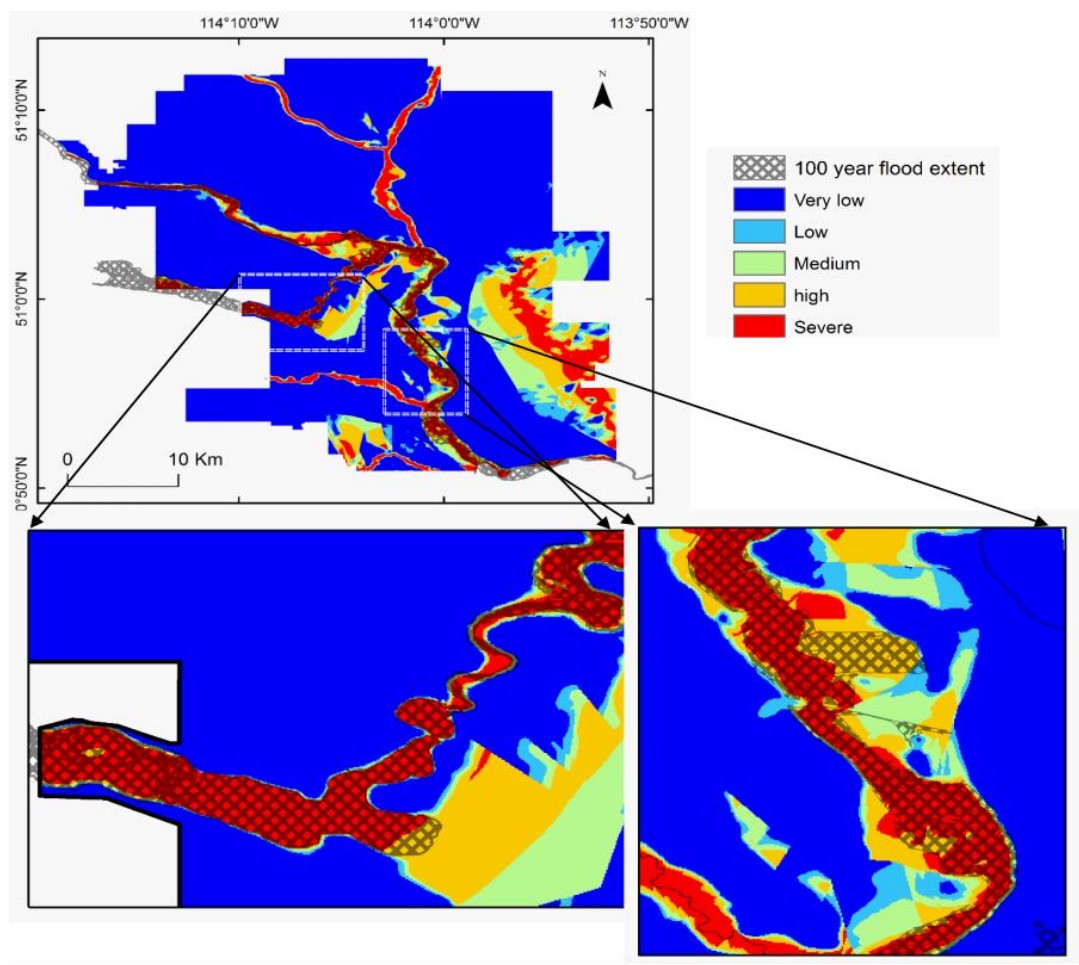

2  Figure 5: Comparison of Hazard map obtained from the present study and a 100-yr flooding
3  extent map prepared by the city of Calgary (Hatched portions). Portions of the reach along the
4  Bow and Elbow rivers are enlarged to show the level of agreement between both maps.





Figure 6. Hazard map for the Red River in Manitoba: (a) without considering flood protection structures in delineating hazard zones and (b) considering flood protection.



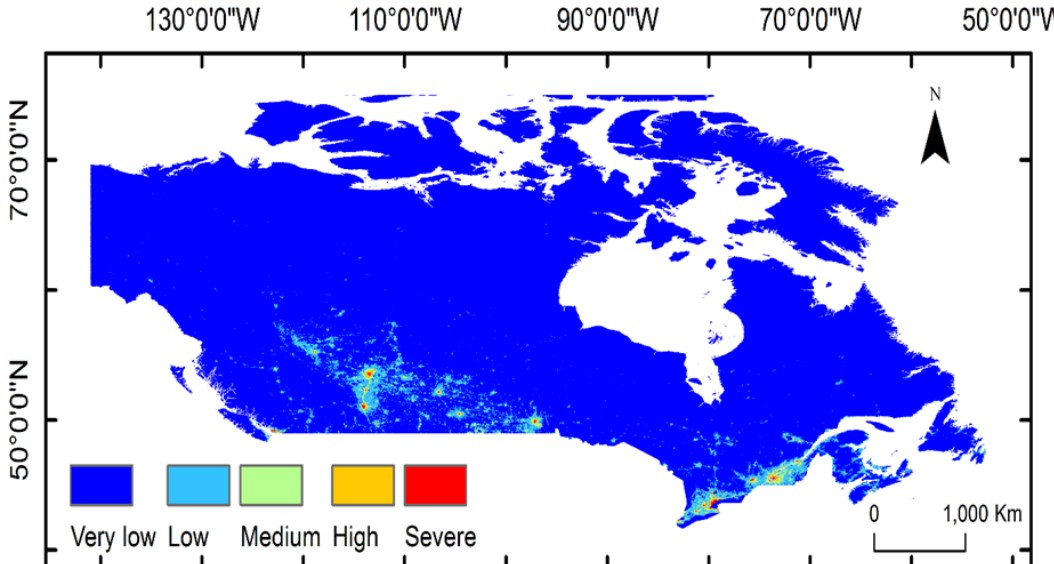

Figure 7: Classified Flood Exposure Map for Canada. Severe and high exposure are concentrated around urban centers in the southern part of the country.



Figure 8. Flood risk map for Canada. Certain portions of the map are enlarged for better visual
interpretation of the various levels of flood risk. The risk map is a product of hazard and
exposure (flood protection measures are not included). Severe risk only occurs in areas of sever
hazard and exposure, causing sever flood risk areas to be concentrated in urban centers.





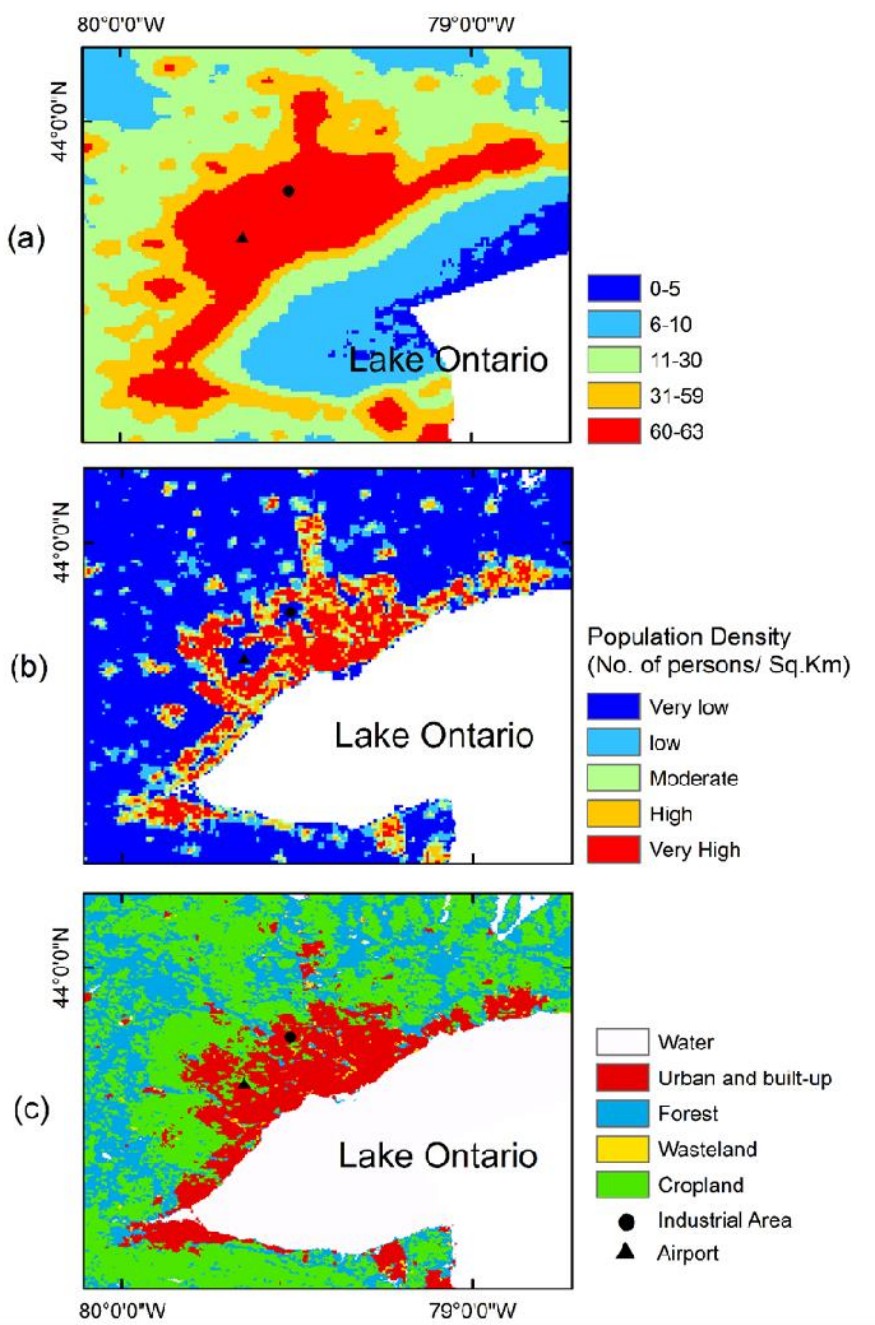

2    Figure 9: Comparison of population distribution and nightlights over Greater Toronto Area: (a)
3    classified nightlights for the area with locations of the airport and a major industrial area; (b)
4    population density over the area; and (c) land-use map of the area indicating urban extent.





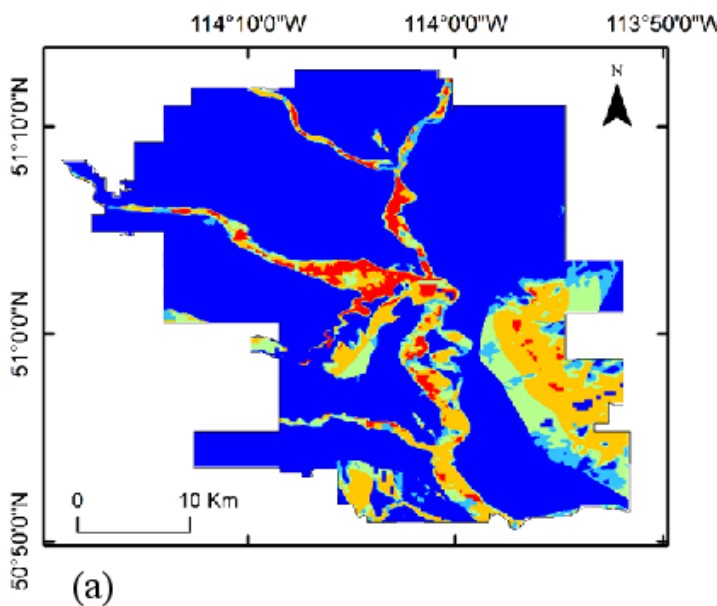

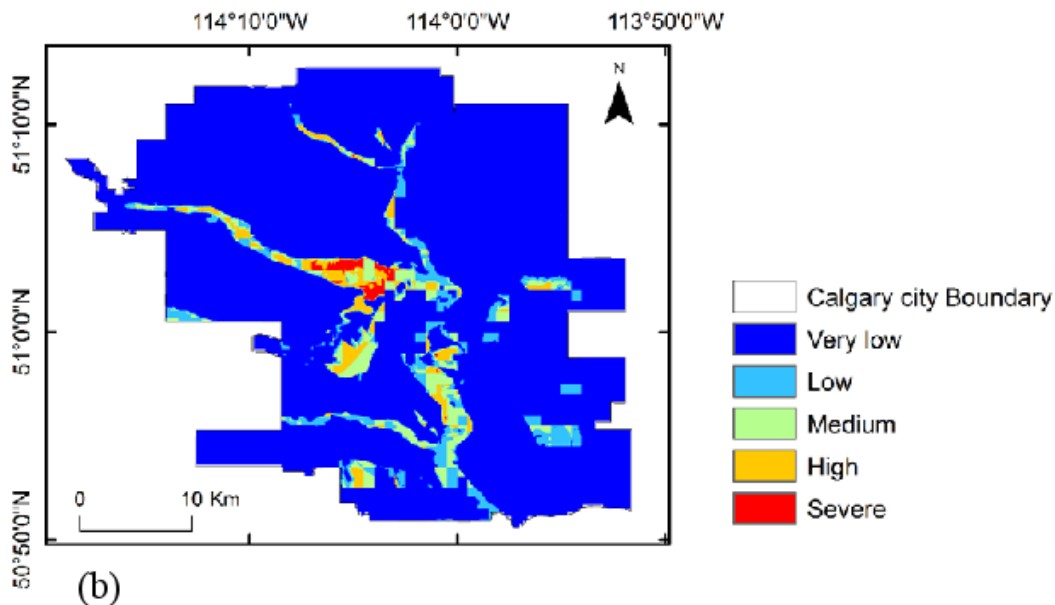

Figure 10: Flood risk map of Calgary; (a) without population information, and (b) with
population. Areas around the center of the city with high rises and dense population remain in
the severe risk category, whereas northern and southern parts, which are mainly commercial,
change to reduced levels of social risk.