# Peer review of "Topography- and nightlight-based national flood risk assessment in Canada"

_Hydrology and Earth System Sciences, 2016_

## Referee Comment (RC1) · Anonymous Referee #1 · 16 Nov 2016

This article provides a series of spatially distributed products for Canada that include a flood hazard map, a flood exposure map and a flood risk map. The indices for each are essentially derived from topography and an indication of population through satellite maps of night lights. The product is assessed by comparison to a 100 year flood plain map of a small region of southern Alberta and to a region around the City of Winnipeg, Manitoba in an effort to relate the product to flood protection infrastructure. The document is well written with few typographical errors so it is easy to follow. There are some good qualities to this paper but I feel there are a few issues that warrant further discussion; in particular the lack of detail in various sections, the lack of discussion and the lack of rigorous validation. Thus, my discussion is more philosophical and about the approach used. My comments are not in any particular order but all speak to these issues to some degree.

[Figure]

1. The authors have used what they consider to be a static entity like topography through two quantities "elevation above nearest drainage" and "distance from nearest drainage" to create a flood hazard level for each grid cell. The floods in the Bow and Elbow Rivers in Calgary, Alberta in 2007 for example, (one of the locations the authors use to verify one of the maps) significantly affected drainage to the point that it changed the rivers' locations, meander and moved a significant amount of sediment. While this would not likely affect a product that is based on a resolution of over 300 metres (at best) because these rivers may not change bank locations by more than 100 metres in one flood, it does beg the question of how often should this product be updated, maintained, etc. Products like this should be given technical support but there is no suggestion of technical support. This is fine because I don't think the development of a product is something that is suitable for publication in HESS and perhaps the authors are more interested in providing an approach leading to a potential product. Well in that case, a much more rigorous evaluation of that approach is required and that is lacking here. What is currently presented is really nothing more than a simple GIS exercise, which I might suggest is not suitable for HESS and thus, the work needs greater discussion, validation and verification if the ultimate objective is indeed to suggest an approach.

2. Page 7 lines 12-19 – The authors need to state in greater detail what they are doing with the comparison around the City of Calgary. Is this a validation or verification? It seems like none of these, than what is this comparison for? If you want to make a comparison, it should be quantitative, instead it is entirely qualitative.

3. Page 8 – The Canada DEM resolution is reported as 326 metres. This is the spatial resolution – what is the elevation resolution and accuracy – 1 metre? 50 cm? What are the implications of this error on flood risk or hazard? The authors combine two topographic indices to create a skewed topographic index and call this flood hazard. I don't necessarily agree that this is flood hazard – what it definitely is, is a new topographic index related to position from a "drainage point". If the authors want to suggest

a surrogate for flood hazard that is easy to create, then they would have to verify that surrogate but that has not been conducted here. At this point, the authors should be true to what they have presented and not label that products as flood hazard but simply the product of two topographically related indices.

4. Page 9 – the authors state "horizontal distance" from nearest drainage network. What is this exactly? Are the authors referring to a buffer like distance? If so, why not just create a buffer? A "horizontal distance" makes no sense in a GIS context, the authors must be careful with their terminology and provide greater detail. For example, in the definition of EAND, the authors intention I suspect is the nearest drainage cell, or point on the drainage network defined by the ArcGIS. But if a point is equally distant from two drainage points, how is the choice made? Details like this should be noted as well as metadata information, errors in the data, etc.

5. Page 9 – line 2 – the authors state that they developed a drainage network as the river network from the ARCGIS tools. Even with a filled DEM, etc, as the authors report, it is well known that a river network derived from a topographic map can often deviate from the actual river network because of errors in the DEM. Given the scale of the DEM used and the size of many of the rivers in Canada, it is possible for drainage points on the DEM derived drainage network not to coincide with actual river locations. Surely this is a problem so why wouldn't the authors use the actual river network for Canada or at least correct their product for actual rivers?

6. One of the reasons why the authors went with such a resolution was because they felt that it made the problem tractable but with "reasonable" detail. But because of the large expanse of this country with little population, there are large areas of the maps with no interest because there are no urban areas. Page 12 refers to Table 2, which shows that the percentage of Canada covered with land use 4 and 5 is less than 6%. The nightlights confirm the enormous area with little population and therefore, with little interest in products like this. It makes me wonder why the authors would create a product that covers all of Canada. Why not create a higher resolution produce that

just focuses on urban areas and simply cut out all the rest? The authors state how problematic political borders are to watershed management. Well then why not create products in only the most hazardous areas? Why not eliminate all the region that is of no interest and not display them? Instead we get maps of the entire extent which has a lot of information that does not have to be displayed or provided. Because the authors rely on visual representation of their work, these visual representations are all that can be critiqued.

7. In Table 3, the percentage of areas covered by high and very high luminosity is tiny in comparison to the rest of the country. The nightlight DN value between 0 and 63 with resolution of one is now descritized into five classes each separated with the same value – one. The authors lump DN values from 11 to 63 for medium to very high luminosity in three out of five classes. Why not instead descretize those regions of interest (medium to very high) into five classes because ultimately you create a skewed product (when you multiply this five level classification with another five level classification scheme) that ignores the detailed information (nightlight, population, land use) and distribution that resides within the two most important classes. In doing this, the authors relegate two whole classes out of five for the bulk of the country that is of no interest. It would make more sense for the authors to focus in on the regions of interest and have five maybe 10 levels of classification within areas of interest. Why did the authors choose five levels of classification and not two, or four or 10?

8. The risk product combines a 326 metre resolution DEM with a 30 arc second DEM. At the Canadian-US border this resolution is probably around 600 metres. So what merging algorithm did the authors use when combining two grids of differing resolutions? What is the ultimate resolution of their product?

9. Page 13 lines 14 - 15, the authors state that "airports and industrial and commercial areas are highly luminous but the census data show low or no population". Floods create numerous environmental hazards that are equally as lethal as is the potential for floods to drown people. If that is what the flood exposure map is about – human harm,

then I would argue, it is incorrect to negate the potential human health risk associated with flood waters having moved through an industrial site simply because no one is living there at night. Flood waters in urban areas are more polluted than sewage and carry harmful hazardous waste that can be extremely harmful if people are exposed. The authors ignore this and simply acknowledge residential areas. This is the general problem I have with this approach.

10. There are too many figures and few that are actually useful. Figure 1 really is not very useful. If you really want to use up valuable journal paper space then why not superimpose (a) and (b)? I would just remove (a).

11. I would appreciate better attention to semantics. For example, on line 13 page 6. How is sufficient defined here by Apel or the authors?

12. Page 14 refers to Figure 2. Again (a) and (b) are both not necessary – just have (b). Figure 3's caption should be revised to read "resulting from EAND X DFND" because this is not a flood hazard map but a map of that index. The topographic index defined by the authors contributes to one kind of flooding but there are others that are equally as hazardous that are not well represented. British Columbia suffers from severe flash flooding that moves enormous amounts of debris yet there seem to be few hazards associated with this type of flooding that is mostly in mountanous regimes showing up in the map because of the way the authors have chosen their index. Can the authors comment on the universality of their choice in Canada? The authors clearly state early in their paper that extreme flooding in Canada is the result of many factors like ice jams, etc. This is very true and thus, the index defined by the authors cannot in fact be toted as a flood hazard by virtue of the fact that what leads to sudden high streamflow – the really danger - is not simply a flat area close to a stream bank. But if that's what the authors want to create, that's okay but then it requires a good discussion of why the approach is novel for defining a flood plain and what the benefits are (like computational ease), then they need to report the computational cost of creating these maps and report a quantitative comparison with things like the 1/100 year flood plain

map in Calgary. Figure 5 refereed to on page 15 shows areas of overlap between the product and the flood plain map. This is again qualitative. A more quantitative comparison is required with even something simply like number of grid cells overlapped versus not overlapped to start with.

13. This brings me to my next point. Large municipal urban centres already have information on high flood risk regions. What information does this product bring them that they don't already have at a better resolution? Risk of fire is largely a problem when it starts encroaching on an urban area and not generally at the same time as a flood risk so how can this low resolution product be helpful to Calgary?

14. The discussion is lacking in many regards in this paper particularly where figures are produced. Page 16 for example refers to figure 6 but honestly, there is nothing really discussed or noted of significance here. Figure 7 is too coarse a resolution to be useful. Figure 8 is an "enlarged" version of an area for better visual interpretation but if they don't provide the exact area in space (not just with hatchmarks but perhaps with an areal photo showing the flood plain in the area) it is not a useful figure. This figure also has little discussion.

15. The authors don't provide a rigorous enough evaluation of their product at this stage. In Figure 10, the authors refer to reduced levels of social risk for commercial regions. Again I disagree with this but perhaps this is due to a lack of rigerous definitions on the part of the authors as to what is "social" – human residential impairment? The authors should revise all their captions to state what is truly shown. Also, there were numerous areal photos of flooded regions within Calgary during the 2013 floods. Why not use this valuable information to compare to their product? That would be a much better evaluation and would demonstrate the deficiencies and limitations of the product in an actual flood that was not 1 in 100 but with an extent that was outside the 1/100 year flood plain.

16. Page 17: line 17, the authors refer to the "average" effect. Why would they be

integrated in the first place? Why is "average" in quotes? My point is that this work is really a GIS exercise and the GIS community understands the issues and limitations with combining data of different resolutions, etc., yet I'm concerned with the lack of attention to terminology or basic GIS concepts used in the discussion. A more formal language is preferred along with greater detail on what was actually created and how.

17. I really do think products like these are good ideas but it's not just what is novel that must be shown but how it is useful and why it is needed. Unfortunately, I do not feel that the reader is given a full understanding of how this approach or product is useful. There is some attempt but more depth is needed. For example, on page 18, line 15, the authors state: "In other regions, and depending on the topography, the 100 year flood might cover two or three of the flood hazard classes." I don't mean to sound curt but so what? How is this useful to a planner that is required by most by-laws to deal with the 100 year flood or design with the 5, 10 or 30 year flood in Canada?

Typographical errors: Line 13, Page 6 – insert "data" after "remotely sensed" Page 8 – insert "an" or "the" before "eight" Page 11 – line 9 replace "from" with "for" Page 32: Spelling eerror in the caption of Figure 8 (should be severe not sever)

---

## Referee Comment (RC2) · Anonymous Referee #2 · 22 Nov 2016

The study aims to develop a national flood risk assessment in Canada by producing a flood hazard map, a flood exposure map, and an economic flood risk map based on global and national spatially distributed data, such as a national DEM, land-use map, nightlight data, and population density information. The flood hazard map is tested against a local inundation map produced by hydraulic modeling for the city of Calgary. The authors also test the influence of flood protection measures on the flood hazard map for the city of Wyoming. The article is well written and easy to follow. It has some interesting aspects, but there are several concerns that will be discussed below.

I miss a clear statement of the research problem and what is novel with the purposed study. The structure of section one and two could be improved by avoiding jumping back and forth between topics.

[Page 8-9] To create the EAND and DFND classes, a drainage network was created using ArcGIS hydrology tool on a coarse resolution DEM. This can produce many errors - why not use an already existing drainage network, or at least verify against one?

[Page 9, Lines 12-13] The classification process for the different maps produced is not clear. For example, the hazard class intervals were selected somewhat arbitrarily. I would like to see more thought behind this, e.g., do they represent floodplains, and why five classes?

[Page 12, Lines 6-8] The exposure map based on nightlight data indicate that 98% of Canada's area has absent or low human activity. This leads to the following question – is a national flood risk assessment useful?

[Page 12, Table 2 and 3; Page 31, Figure 7; Page 32, Figure 8] The land-use classes and the nightlight classification used for the exposure map give northern communities very low or low exposure level by default, resulting in very low or low flood exposure, and very low flood risk in areas above 60° N. Is this national flood risk map useful for residents above 60° N? I am missing a discussion around how the classification process affects the end product.

[Page 14-15, Lines 14-21, 1-5] A coarser DEM is chosen for the study to keep computational costs low, but results show that a finer resolution DEM (20 m in this case) gives better results and a more reliable flood hazard assessment. Floods are usually analyzed and managed at the provincial level in Canada where local information is important, why is a national flood risk assessment needed?

[Page 15, Lines 16-20] It is suggested that hazard levels can be reclassified locally to match floods with different return periods in areas where flood inundation using hydraulic modeling is available. But, how useful are local topography-based flood hazard maps where flood inundation maps based on hydraulic modeling already exist? Also, topography-based flood hazard maps does not account for backwater and other hydraulic effects on areas upstream of flood protection. One related question is also how

useful flood hazard maps with different return periods are if many floods are caused by ice-jams [Page 7, Lines 7-8; Page 18, Lines 11-14]?

[Page 16-17, Lines 23-24, 1-3] The authors bring up the issue with overglow effect when analyzing nightlight data. Have potential overglow effects been analyzed for the 2013 nightlight data used in this study, e.g., in comparison with previous years?

[Page 17, Lines 10-19] There is a discussion that population data should be used together with nightlight data to separate social and economic impact, as airports and industrial areas show high luminous values but low population density. I will argue that although these built-up areas have low population density, they have high social impact, e.g., airports.

[Page 19, Lines 12-16] There are many uncertainty aspects with the classes identified and some of the methods used – is the final product really useful and practical [Page 20, Lines 6-7] - also when considering the shortcomings the authors have presented?

The article has 10 figures, are all of them needed? For example, Figure 1 – a and b should be combined if to be included at all. Also, is both a and b in Figure 2 needed, they show the same information. Figure 5 – exclude enlarged figures, and visually improve the main figure.

Minor issues: [Page 1, Line 13] The authors state that the study uses datasets at reasonably fine resolutions to create flood risk maps – what is considered reasonable?

[Page 9, Line 4] What do you mean by horizontal distance?

[Page 9, Line7] EAND instead of EFND

[Page 11, Lines 19-22] It is stated that the average values of all nightlight satellites were used in this study, but there is only one available for 2013.

[Page 17, Line 17] What is the "average" effect?

[Page 21, Line 31] De Moel should be de Moel
[Page 23, Line 25] The reference Schanze is not found in the text

---

## Author Comment (AC1) · 24 Nov 2016

The authors would like to thank the anonymous reviewer for providing a very thoughtful assessment and very useful suggestions. We are providing here below our detailed response to each remark. 1. The authors have used what they consider to be a static entity like topography through two quantities "elevation above nearest drainage" and "distance from nearest drainage" to create a flood hazard level for each grid cell. The floods in the Bow and Elbow Rivers in Calgary, Alberta in 2007 for example, (one of the locations the authors use to verify one of the maps) significantly affected drainage to the point that it changed the rivers' locations, meander and moved a significant amount of sediment. While this would not likely affect a product that is based on a resolution of over 300 metres (at best) because these rivers may not change bank locations by more than 100 metres in one flood, it does beg the question of how often

should this product be updated, maintained, etc. Products like this should be given technical support but there is no suggestion of technical support. This is fine because I don't think the development of a product is something that is suitable for publication in HESS and perhaps the authors are more interested in providing an approach leading to a potential product. Well in that case, a much more rigorous evaluation of that approach is required and that is lacking here. What is currently presented is really nothing more than a simple GIS exercise, which I might suggest is not suitable for HESS and thus, the work needs greater discussion, validation and verification if the ultimate objective is indeed to suggest an approach.

R1. We would like to emphasize that the approach we are proposing here proposes for the first time the integration of detailed topographic information, in the form of distance and elevation from streams, with hydrologic and human settlements information to assess flood risk. What is obtained here is much more than a flood inundation map, as we integrate information on hazard and exposure, therefore moving a step forward towards large scale estimation of flood risk. Actually, what is intended here is both an approach that can be followed in any place across the globe and a product (for Canada). Therefore we believe that the article is presenting significant innovation. For example, many developing countries can benefit from this as global remotely sensed data are becoming increasingly available. We validated the approach using the example of Calgary in a qualitative visual way, but in the revised manuscript we will also provide the evaluation quantitatively in the form of error metrics. We agree that big floods may change the river course, however, even most local hydraulic modeling for flood inundation purposes do not consider such geomorphological changes. Our approach can be easily updated when significant topographical changes happen in the landscape and this information is updated into the DEM being used. We believe little technical support is needed as we can provide relevant codes and GIS layers that can be re-run when significant changes happen in topography or landuse.

2. Page 7 lines 12-19 – The authors need to state in greater detail what they are doing

with the comparison around the City of Calgary. Is this a validation or verification? It seems like none of these, than what is this comparison for? If you want to make a comparison, it should be quantitative, instead it is entirely qualitative.

R2. It would be useful if the reviewer clarified what is meant by validation and verification, as these terms are sometimes used in hydrology with different meanings with respect to what is defined, for instance, in the ISO 9000 rule (for more details please see https://en.wikipedia.org/wiki/Verification_and_validation; see also Biondi et al., 2012). Our application to the city of Calgary is intended to be a validation, according to the following definition of the term: "Validation is the assurance that a product, service, or system meets the needs of the customer and other identified stakeholders. It often involves acceptance and suitability with external customers". To meet the above requirements, in hydrology validation is often performed by referring to independent set of data, as we did in our case. We agree with the reviewer that it is possible to provide a more quantitative assessment of the results. As mentioned earlier, in the revised version of the paper we will indeed make our validation quantitative, and will call it evaluation of the topography-based flood hazard mapping.

3. Page 8 – The Canada DEM resolution is reported as 326 metres. This is the spatial resolution – what is the elevation resolution and accuracy – 1 metre? 50 cm? What are the implications of this error on flood risk or hazard? The authors combine two topographic indices to create a skewed topographic index and call this flood hazard. I don't necessarily agree that this is flood hazard – what it definitely is, is a new topographic index related to position from a "drainage point". If the authors want to suggest a surrogate for flood hazard that is easy to create, then they would have to verify that surrogate but that has not been conducted here. At this point, the authors should be true to what they have presented and not label that products as flood hazard but simply the product of two topographically related indices.

R3. We respectfully disagree. There is no universal measure of flood hazard. Typically probability of occurrence is used. Here we are assuming that our proposed classification of the landscape, in the surrounding of the rivers, based on topography reflects its probability of being flooded, and thus, reflects hazard. We will make the assumption clearer. We are currently reproducing the entire work using a 20m resolution DEM that has a vertical accuracy ranging from zero to 10m for more than 90% of the entire country (Natural resources Canada, 2013; Beaulieu and Clavet, 2009). Information on metadata and errors will be provided at relevant locations. Therefore, the reliability of the DEM is not a question and, in general, does not affect the validity of the approach and the assumption that flood hazard can be inferred from landscape topography. Others have related flood hazard maps to topography, e.g. Lugeri et al. (2010), which is cited in our manuscript. As this approach can be followed using any elevation dataset, readers could reproduce these maps with improved accuracy in the presence of more accurate and finer DEMs/DTMs. Statements mentioning the vertical accuracy and its implications on the flood hazard map will also be included in the revised manuscript.

4. Page 9 – the authors state "horizontal distance" from nearest drainage network. What is this exactly? Are the authors referring to a buffer like distance? If so, why not just create a buffer? A "horizontal distance" makes no sense in a GIS context, the authors must be careful with their terminology and provide greater detail. For example, in the definition of EAND, the authors intention I suspect is the nearest drainage cell, or point on the drainage network defined by the ArcGIS. But if a point is equally distant from two drainage points, how is the choice made? Details like this should be noted as well as metadata information, errors in the data, etc.

R4. We are referring to a buffer like distance while describing DFND. However, in GIS, the term "buffer" is usually applied to concentric distances to a feature (line, point or polygon) in vector format. For the present study, the stream network was retained in raster format to maintain consistency in all subsequent calculations. Horizontal distance refers to the Euclidean distance between the drainage cells and adjoining cells that are estimated using the "Euclidean distance" tool in ArcGIS, followed by reclassification using the limits mentioned in Table 1. Hence, the word "buffer" was avoided

and "horizontal distance" used instead. The term horizontal was used as this measure considers only the distance and not the elevation difference between the drainage cells and the adjoining cells. The reviewer is right that in EAND, the elevations to the nearest drainage cell is estimated as described in section 3.1. Additional metadata information on the DEM and errors, as well additional details to clarify the procedure, will be included in the revised manuscript.

5. Page 9 – line 2 – the authors state that they developed a drainage network as the river network from the ARCGIS tools. Even with a filled DEM, etc, as the authors report, it is well known that a river network derived from a topographic map can often deviate from the actual river network because of errors in the DEM. Given the scale of the DEM used and the size of many of the rivers in Canada, it is possible for drainage points on the DEM derived drainage network not to coincide with actual river locations. Surely this is a problem so why wouldn't the authors use the actual river network for Canada or at least correct their product for actual rivers?

R5. Some of the Reviewer's concerns will be addressed when we present everything using the 20 m DEM. Even the river network made available through Environment and Climate Change Canada (ECCC) is generated using DEMs, and seems to be based on even coarser DEM than 20 m. We used Google Earth to compare the river network we generated against actual rivers, and the comparison, which validates our approach, will be presented in our Response letter.

6. One of the reasons why the authors went with such a resolution was because they felt that it made the problem tractable but with "reasonable" detail. But because of the large expanse of this country with little population, there are large areas of the maps with no interest because there are no urban areas. Page 12 refers to Table 2, which shows that the percentage of Canada covered with land use 4 and 5 is less than 6%. The nightlights confirm the enormous area with little population and therefore, with little interest in products like this. It makes me wonder why the authors would create a product that covers all of Canada. Why not create a higher resolution produce that

just focuses on urban areas and simply cut out all the rest? The authors state how problematic political borders are to watershed management. Well then why not create products in only the most hazardous areas? Why not eliminate all the region that is of no interest and not display them? Instead we get maps of the entire extent which has a lot of information that does not have to be displayed or provided. Because the authors rely on visual representation of their work, these visual representations are all that can be critiqued.

R6. We respectfully disagree with the Reviewer as this suggestion contradicts the purpose of our work. Several municipalities attempt to model or use consultants to hydraulically model the few kilometers river reaches that pass through urban areas, but the bigger picture of an entire province or Basin is missing. Development of new areas is moving fast in Canada and encroaching into flood hazard areas is happening (as we presented the case of Fort McMurray) because such areas were not modeled as they were not populated! The flood hazard map indicates that larger areas of Canada are in significantly high flood hazard areas, but the vulnerability is, of course, centered around urban areas. There is a difference, and we need to highlight hazard areas to help planning and future developments, and also indicate the flood hazards in agricultural areas, important heritage areas, vulnerable ecosystems,..etc. It is not just about urban centres.

7. In Table 3, the percentage of areas covered by high and very high luminosity is tiny in comparison to the rest of the country. The nightlight DN value between 0 and 63 with resolution of one is now descritized into five classes each separated with the same value – one. The authors lump DN values from 11 to 63 for medium to very high luminosity in three out of five classes. Why not instead descretize those regions of interest (medium to very high) into five classes because ultimately you create a skewed product (when you multiply this five level classification with another five level classification scheme) that ignores the detailed information (nightlight, population, land use) and distribution that resides within the two most important classes. In doing this,

the authors relegate two whole classes out of five for the bulk of the country that is of no interest. It would make more sense for the authors to focus in on the regions of interest and have five maybe 10 levels of classification within areas of interest. Why did the authors choose five levels of classification and not two, or four or 10?

R7. We believe this is related to the earlier point of focusing on smaller urbanized part of Canada or doing the entire country. Our choice and preference is the latter, but other researchers are free to take our approach and focus on any area they prefer.

8. The risk product combines a 326 metre resolution DEM with a 30 arc second DEM. At the Canadian-US border this resolution is probably around 600 metres. So what merging algorithm did the authors use when combining two grids of differing resolutions? What is the ultimate resolution of their product?

R8. Only one DEM (326m resolution) was used in the preparation of risk map for the entire country. The 30 arc second resolution corresponds to the Nightlight dataset that was used to prepare the exposure map. The nightlight images were resampled to match the resolution of the DEM within the entire study area and the final risk map was produced by combining the hazard and exposure map. The final product was a risk map of 326 m resolution itself. For further clarity, a description on the resolution of the derived maps will be provided in the revised manuscript. 9. Page 13 lines 14 - 15, the authors state that "airports and industrial and commercial areas are highly luminous but the census data show low or no population". Floods create numerous environmental hazards that are equally as lethal as is the potential for floods to drown people. If that is what the flood exposure map is about – human harm, then I would argue, it is incorrect to negate the potential human health risk associated with flood waters having moved through an industrial site simply because no one is living there at night. Flood waters in urban areas are more polluted than sewage and carry harmful hazardous waste that can be extremely harmful if people are exposed. The authors ignore this and simply acknowledge residential areas. This is the general problem I have with this approach.

R9. We agree, we just wanted to show that census data showing zero population do not mean no human presence. There is still capital investment. Human lives are disturbed at a different level when homes are impacted, more than having a place of work impacted, but certainly human harm could still happen in industrial areas. We think that a statement about this in the revised manuscript will address the issue.

10. There are too many figures and few that are actually useful. Figure 1 really is not very useful. If you really want to use up valuable journal paper space then why not superimpose (a) and (b)? I would just remove (a).

R10. Sure, this can be done.

11. I would appreciate better attention to semantics. For example, on line 13 page 6. How is sufficient defined here by Apel or the authors?

R11. Sure, it is subjective term that relates to "acceptable" level of accuracy and representation. Different users and uses dictate different levels of acceptability.

12. Page 14 refers to Figure 2. Again (a) and (b) are both not necessary – just have (b). Figure 3's caption should be revised to read "resulting from EAND X DFND" because this is not a flood hazard map but a map of that index. The topographic index defined by the authors contributes to one kind of flooding but there are others that are equally as hazardous that are not well represented. British Columbia suffers from severe flash flooding that moves enormous amounts of debris yet there seem to be few hazards associated with this type of flooding that is mostly in mountanous regimes showing up in the map because of the way the authors have chosen their index. Can the authors comment on the universality of their choice in Canada? The authors clearly state early in their paper that extreme flooding in Canada is the result of many factors like ice jams, etc. This is very true and thus, the index defined by the authors cannot in fact be toted as a flood hazard by virtue of the fact that what leads to sudden high streamflow – the really danger - is not simply a flat area close to a stream bank. But if that's what the authors want to create, that's okay but then it requires a good discussion of

why the approach is novel for defining a flood plain and what the benefits are (like computational ease), then they need to report the computational cost of creating these maps and report a quantitative comparison with things like the 1/100 year flood plain map in Calgary. Figure 5 refereed to on page 15 shows areas of overlap between the product and the flood plain map. This is again qualitative. A more quantitative comparison is required with even something simply like number of grid cells overlapped versus not overlapped to start with.

R12. Figure 2 (a) and (b) help the readers see the difference that reclassification into 5 classes cause to the map. As discussed earlier, we disagree on the issue of Hazard definition. The issue of debris from the mountain can be just another index added to Hazard based on proximity to erodible mountains in the headwaters. The issue of ice jamming is true as a cause of flooding, but inundated areas first impacted are the lowlands and lands close to the stream, we cannot see how ice jams negates the universality of our proposed hazard index.

13. This brings me to my next point. Large municipal urban centres already have information on high flood risk regions. What information does this product bring them that they don't already have at a better resolution? Risk of fire is largely a problem when it starts encroaching on an urban area and not generally at the same time as a flood risk so how can this low resolution product be helpful to Calgary?

R13. We believe that we addressed this point earlier, and also in the manuscript. What an approach like this brings is different. This approach helps prioritize areas for detailed modeling, help development planning, and other studies such as investigation various population groups and their vulnerability to certain hazards, which is useful for resource allocation.

14. The discussion is lacking in many regards in this paper particularly where figures are produced. Page 16 for example refers to figure 6 but honestly, there is nothing really discussed or noted of significance here. Figure 7 is too coarse a resolution to be

useful. Figure 8 is an "enlarged" version of an area for better visual interpretation but if they don't provide the exact area in space (not just with hatchmarks but perhaps with an areal photo showing the flood plain in the area) it is not a useful figure. This figure also has little discussion.

R14. We believe that enough discussion is provided for Figure 6 and others. However, we will attempt to expand on figure discussion a bit in the revised manuscript.

15. The authors don't provide a rigorous enough evaluation of their product at this stage. In Figure 10, the authors refer to reduced levels of social risk for commercial regions. Again I disagree with this but perhaps this is due to a lack of rigerous definitions on the part of the authors as to what is "social" – human residential impairment? The authors should revise all their captions to state what is truly shown. Also, there were numerous areal photos of flooded regions within Calgary during the 2013 floods. Why not use this valuable information to compare to their product? That would be a much better evaluation and would demonstrate the deficiencies and limitations of the product in an actual flood that was not 1 in 100 but with an extent that was outside the 1/100 year flood plain.

R15. The issue of population in residential and commercial/industrial areas was discussed earlier. We compared with 100-year flood modeling in Calgary because a georeferenced map was made available to us, and the hydraulic modeling is supposed to be the "accurate and scientific" way of mapping flood inundation. Therefore, enough information is available to perform a validation of our results. However, we also managed to obtain a high resolution aerial phot of the 2013 flood in the Qu'Appelle River Basin, which is one of the most challenging areas in Canada for hydrologic and hydraulic modeling – the Canadian Prairies. We compared our product with this photo and the results are very good, and we will show this in our response letter and perhaps in the revised manuscript. We thank the reviewer for bringing this up because it gave us the opportunity to conduct another compelling validation of our approach and product.

16. Page 17: line 17, the authors refer to the "average" effect. Why would they be integrated in the first place? Why is "average" in quotes? My point is that this work is really a GIS exercise and the GIS community understands the issues and limitations with combining data of different resolutions, etc., yet I'm concerned with the lack of attention to terminology or basic GIS concepts used in the discussion. A more formal language is preferred along with greater detail on what was actually created and how.

R16. Simply what we meant is that integrating two aspects in one can mask the individual effects. Sometimes integration is a must, and we did it, for example, with EAND and DFND, but in case of effect on population we wanted it to be explicit, that's all! As suggested, additional details on the GIS-related analysis will be provided at relevant locations in the results and discussion sections in the revised manuscript. We agree that the application of the work is basically a GIS exercise, which however makes use of an innovative idea.

17. I really do think products like these are good ideas but it's not just what is novel that must be shown but how it is useful and why it is needed. Unfortunately, I do not feel that the reader is given a full understanding of how this approach or product is useful. There is some attempt but more depth is needed. For example, on page 18, line 15, the authors state: "In other regions, and depending on the topography, the 100 year flood might cover two or three of the flood hazard classes." I don't mean to sound curt but so what? How is this useful to a planner that is required by most by-laws to deal with the 100 year flood or design with the 5, 10 or 30 year flood in Canada? Typographical errors: Line 13, Page 6 – insert "data" after "remotely sensed" Page 8 – insert "an" or "the" before "eight" Page 11 – line 9 replace "from" with "for" Page 32: Spelling eerror in the caption of Figure 8 (should be severe not sever)

R17. We believe that our approach is indeed presenting an original contribution, and we also believe that it is extremely useful. It allows the identification of critical areas, where subsequent detailed analyses should focus on. For example, local authorities may want relate flows at different flood frequencies (e.g., 100 year) to water stage (can

be done using rating curves available locally). The stage of different floods will indicate clearly which of our hazard classes will be inundated. This way local authorities can convert our map to flood frequencies. We will better explain in the revised manuscript the practical advantages that an extensive and quick mapping of risk may provide for local management and decision making.

References

Beaulieu, A., & Clavet, D. (2009). Accuracy assessment of Canadian digital elevation data using ICESat. Photogrammetric Engineering & Remote Sensing, 75(1), 81-86.

Biondi, D., Freni, G., Iacobellis, V., Mascaro, G., Montanari, A., (2012), Validation of hydrological models: Conceptual basis, methodological approaches and a proposal for a code of practice, Physics and Chemistry of the Earth, 42-44, 70-76.

Natural Resources Canada (2013) Canadian Digital Elevation Model Product Specifications- Edition 1.1, Government of Canada, pp 11.

Lugeri, N., Z. Kundzewicz, E. Genovese, S. Hochrainer, and M. Radziejewski (2010), River flood 33 risk and adaptation in Europe – assessment of the present status, Mitig. Adapt. Strateg. Glob. 34 Change, 15, 621–639.

---

## Author Response (AR1)

Response letter to the Reviewers of "Topography- and nightlight-based national flood risk
assessment in Canada" – HESS-2016-524.
Reviewer 1
The authors would like to thank the anonymous reviewer #1 for providing a very thoughtful
assessment and very useful suggestions. We are providing here below our detailed response to
each remark.
*1.* *The authors have used what they consider to be a static entity like topography through two*
*quantities "elevation above nearest drainage" and "distance from nearest drainage" to create a*
*flood hazard level for each grid cell. The floods in the Bow and Elbow Rivers in Calgary, Alberta*
*in 2007 for example, (one of the locations the authors use to verify one of the maps) significantly*
*affected drainage to the point that it changed the rivers' locations, meander and moved a*
*significant amount of sediment. While this would not likely affect a product that is based on a*
*resolution of over 300 metres (at best) because these rivers may not change bank locations by*
*more than 100 metres in one flood, it does beg the question of how often should this product be*
*updated, maintained, etc. Products like this should be given technical support but there is no*
*suggestion of technical support. This is fine because I don't think the development of a product is*
*something that is suitable for publication in HESS and perhaps the authors are more interested*
*in providing an approach leading to a potential product. Well in that case, a much more rigorous*
*evaluation of that approach is required and that is lacking here. What is currently presented is*
*really nothing more than a simple GIS exercise, which I might suggest is not suitable for HESS*
*and thus, the work needs greater discussion, validation and verification if the ultimate objective is*
*indeed to suggest an approach.*
**R1.** We would like to emphasize that the approach we are adopting here proposes for the first time
the integration of detailed topographic information, in the form of distance and elevation from
streams, with hydrologic and human settlements information to assess flood risk. What is obtained
here is much more than a flood inundation map, as we integrate information on hazard and
exposure, therefore moving a step forward towards large scale estimation of flood risk. Actually,
what is intended here is both an approach that can be followed in any place across the globe and a
product (for Canada). Therefore we believe that the article is presenting significant innovation. For
example, many developing countries can benefit from this as global remotely sensed data are
becoming increasingly available. We agree with the reviewer that a rigorous evaluation of the
approach is needed. Accordingly, we revised the manuscript (Page 10, line 13 – page 11, line 15)
to make the validation of the hazard map quantitative. We also added another qualitative
assessment (Page 18, lines 13-21 and Figure 5) to compare the flood hazard mapping approach
against an aerial photo showing the actual extent of 2013 flood in the Qu'Appelle river.
We agree that big floods may change the river course and therefore an update of the results from
any hydraulic model may be needed after an extreme event. Actually, our approach can be easily
updated when significant topographical changes happen in the landscape and this information is
updated into the DEM being used. In this regard, updating the product proposed here can be easier than reconducting detailed hydraulic modeling. We believe little technical support is needed as we can provide relevant codes and GIS layers that can be re-run when significant changes happen in topography or landuse.

*2. Page 7 lines 12-19 – The authors need to state in greater detail what they are doing with the comparison around the City of Calgary. Is this a validation or verification? It seems like none of these, than what is this comparison for? If you want to make a comparison, it should be quantitative, instead it is entirely qualitative.*

**R2.** It would be useful if the reviewer clarified what is meant by validation and verification, as these terms are sometimes used in hydrology with different meanings with respect to what is defined, for instance, in the ISO 9000 rule (for more details please see https://en.wikipedia.org/wiki/Verification_and_validation; see also Biondi et al., 2012). Our application to the city of Calgary is intended to be a validation, according to the following definition of the term: "Validation is the assurance that a product, service, or system meets the needs of the customer and other identified stakeholders. It often involves acceptance and suitability with external customers". To meet the above requirements, in hydrology validation is often performed by referring to independent set of data, as we did in our case. We clarified in the revised version of the paper that we are providing a validation according to the above definition.

As mentioned in the previous comment, we already provided a quantitative comparison in the revised manuscript, and also added another qualitative assessment.

*3. Page 8 – The Canada DEM resolution is reported as 326 metres. This is the spatial resolution – what is the elevation resolution and accuracy – 1 metre? 50 cm? What are the implications of this error on flood risk or hazard? The authors combine two topographic indices to create a skewed topographic index and call this flood hazard. I don't necessarily agree that this is flood hazard – what it definitely is, is a new topographic index related to position from a "drainage point". If the authors want to suggest a surrogate for flood hazard that is easy to create, then they would have to verify that surrogate but that has not been conducted here. At this point, the authors should be true to what they have presented and not label that products as flood hazard but simply the product of two topographically related indices.*

**R3.** We respectfully disagree. There is no universal measure of flood hazard. Typically, probability of occurrence is used. Here we are assuming that our proposed classification of the landscape, in the surrounding of the rivers, based on topography reflects its probability of being flooded, and thus, reflects hazard. We reproduced the entire work using DEM-20 that has a vertical accuracy ranging from zero to 10 m for more than 90% of the entire country (Natural resources Canada, 2013; Beaulieu and Clavet, 2009). Information on metadata and errors is now provided in the revised manuscript (Page 8, line 20 – Page 9, line 6). Therefore, the reliability of the DEM is not a question and, in general, does not affect the validity of the approach and the assumption that flood hazard can be inferred from landscape topography. Others have related flood hazard maps to topography, e.g. Lugeri et al. (2010), which is cited in our manuscript. As this approach can be followed using any elevation dataset, readers could reproduce these maps with improved accuracy in the presence of more accurate and finer DEMs/DTMs.

*__4.__ Page 9 – the authors state "horizontal distance" from nearest drainage network. What is this exactly? Are the authors referring to a buffer like distance? If so, why not just create a buffer? A "horizontal distance" makes no sense in a GIS context, the authors must be careful with their terminology and provide greater detail. For example, in the definition of EAND, the authors intention I suspect is the nearest drainage cell, or point on the drainage network defined by the ArcGIS. But if a point is equally distant from two drainage points, how is the choice made? Details like this should be noted as well as metadata information, errors in the data, etc.*

__R4.__ We are referring to a buffer like distance while describing DFND. However, in GIS, the term "buffer" is usually applied to concentric distances to a feature (line, point or polygon) in vector format. For the present study, the stream network was retained in raster format to maintain consistency in all subsequent calculations. Horizontal distance refers to the Euclidean distance between the drainage cells and adjoining cells that are estimated using the "Euclidean distance" tool in ArcGIS, followed by reclassification using the limits mentioned in Table 1. Hence, the word "buffer" was avoided and "horizontal distance" used instead. The term horizontal was used as this measure considers only the distance and not the elevation difference between the drainage cells and the adjoining cells. The reviewer is right that in EAND, the elevations to the nearest drainage cell is estimated as described in section 3.1. Additional metadata information on the DEM and errors, as well additional details to clarify the procedure, is included in the revised manuscript (Page 8, line 20 – Page 9, line 6).

*__5.__ Page 9 – line 2 – the authors state that they developed a drainage network as the river network from the ARCGIS tools. Even with a filled DEM, etc, as the authors report, it is well known that a river network derived from a topographic map can often deviate from the actual river network because of errors in the DEM. Given the scale of the DEM used and the size of many of the rivers in Canada, it is possible for drainage points on the DEM derived drainage network not to coincide with actual river locations. Surely this is a problem so why wouldn't the authors use the actual river network for Canada or at least correct their product for actual rivers?*

__R5.__ Some of the Reviewer's concerns were already addressed as we presented everything using the DEM-20. Even the river network made available through Environment and Climate Change Canada (ECCC) is generated using DEMs, and is based on even coarser DEM than 20 m. To verify this, the stream network delineated using the 20m DEM (Blue lines) and the stream network available from ECCC (Red lines) were overlaid on Satellite imagery available in Google Earth at different locations in Canada. Results for two such locations are presented in the following figures (L1 and L2). It can be clearly seen that the stream network delineated using the 20m follows that actual path of the streams and also captures any meandering whereas the readily available stream network presents only as straight lines cutting through the terrain.

[Figure]

Fig.L1 Comparison of river network obtained from the 20m DEM (Blue lines) with the river network given by ECCC (red lines) in the Greater Toronto Area, ON

[Figure]

Fig.L2 Comparison of river network obtained from the 20m DEM (Blue lines) with the river network given by ECCC (red lines) in Ottawa, QC

*6. One of the reasons why the authors went with such a resolution was because they felt that it made the problem tractable but with "reasonable" detail. But because of the large expanse of this country with little population, there are large areas of the maps with no interest because there are no urban areas. Page 12 refers to Table 2, which shows that the percentage of Canada covered with land use 4 and 5 is less than 6%. The nightlights confirm the enormous area with little population and therefore, with little interest in products like this. It makes me wonder why the authors would create a product that covers all of Canada. Why not create a higher resolution produce that just focuses on urban areas and simply cut out all the rest? The authors state how problematic political borders are to watershed management. Well then why not create products in only the most hazardous areas? Why not eliminate all the region that is of no interest and not display them? Instead we get maps of the entire extent which has a lot of information that does not have to be displayed or provided. Because the authors rely on visual representation of their work, these visual representations are all that can be critiqued.*

**R6.** We respectfully disagree with the Reviewer as this suggestion contradicts the purpose of our work. Several municipalities attempt to model or use consultants to hydraulically model the few kilometers river reaches that pass through urban areas, but the bigger picture of an entire province or basin is missing. Development of new areas is moving fast in Canada and encroachment into flood hazard areas is happening (as we presented the case of Fort McMurray) because such areas were not modeled as they were not populated! The flood hazard map indicates that larger areas of Canada are in significantly high flood hazard areas, but the exposure is, of course, centered around urban areas because of the adopted definition of exposure. We need to highlight hazard areas to help planning and future developments, and also indicate the flood hazards in agricultural areas, important heritage areas, transportation infrastructures, such as roads crossing unurbanized areas, vulnerable ecosystems, etc. It is not just about urban centres. To demonstrate this, we present the figure L3. The figure is for a location north of Edmonton, Alberta, where the exposure map indicates "very low" to "low" exposure. However, a road network map overlaid on top of it indicates that there is a dense road network connecting different locations within the area. The hazard map for the same area also indicates a dense network with hazard levels "severe" and "high". Roads were flooded in major Canadian flood events and hampered rescue efforts.

[Figure]

Figure L3: Exposure (left panel) and hazard (right panel) maps overlaid with road network.

*__7.__ In Table 3, the percentage of areas covered by high and very high luminosity is tiny in comparison to the rest of the country. The nightlight DN value between 0 and 63 with resolution of one is now descritized into five classes each separated with the same value – one. The authors lump DN values from 11 to 63 for medium to very high luminosity in three out of five classes. Why not instead descretize those regions of interest (medium to very high) into five classes because ultimately you create a skewed product (when you multiply this five level classification with another five level classification scheme) that ignores the detailed information (nightlight, population, land use) and distribution that resides within the two most important classes. In doing this, the authors relegate two whole classes out of five for the bulk of the country that is of no interest. It would make more sense for the authors to focus in on the regions of interest and have five maybe 10 levels of classification within areas of interest. Why did the authors choose five levels of classification and not two, or four or 10?*

__R7.__ We believe this is related to the earlier point of focusing on smaller urbanized part of Canada or doing the entire country. Our choice and preference is the latter, but other researchers are free to take our approach and focus on any area they prefer, and also other hazard classifications in number of classes and ranges.

*__8.__ The risk product combines a 326 metre resolution DEM with a 30 arc second DEM. At the Canadian-US border this resolution is probably around 600 metres. So what merging algorithm*

*did the authors use when combining two grids of differing resolutions? What is the ultimate resolution of their product?*

**R8.** In the revised manuscript, DEM-20 (20 m resolution) was used in the preparation of risk map for the entire country. The 30 arc second resolution corresponds to the Nightlight dataset that was used to prepare the exposure map. The nightlight images were resampled to match the resolution of the DEM within the entire study area and the final risk map was produced by combining the hazard and exposure map. The final product was a risk map of 20 m resolution. This clarification has been been added to the revised manuscript (Page 13, lines 18-20).

*__9.__ Page 13 lines 14 - 15, the authors state that "airports and industrial and commercial areas are highly luminous but the census data show low or no population". Floods create numerous environmental hazards that are equally as lethal as is the potential for floods to drown people. If that is what the flood exposure map is about – human harm, then I would argue, it is incorrect to negate the potential human health risk associated with flood waters having moved through an industrial site simply because no one is living there at night. Flood waters in urban areas are more polluted than sewage and carry harmful hazardous waste that can be extremely harmful if people are exposed. The authors ignore this and simply acknowledge residential areas. This is the general problem I have with this approach.*

**R9.** We agree, we just wanted to show that census data showing zero population do not mean no human presence. There is still capital investment. Human lives are disturbed at a different level when homes are impacted, more than having a place of work impacted, but certainly human harm could still happen in industrial areas. We clarified this by a statement on Page 15, lines 17-18.

*__10.__ There are too many figures and few that are actually useful. Figure 1 really is not very useful. If you really want to use up valuable journal paper space then why not superimpose (a) and (b)? I would just remove (a).*

**R10.** Figure 1 is now modified by combining old figures 1(a) and 1(b) and relevant write up is now provided in page 6, Lines 1-7.

*__11.__ I would appreciate better attention to semantics. For example, on line 13 page 6. How is sufficient defined here by Apel or the authors?*

**R11.** It is subjective term that relates to "acceptable" level of accuracy and representation. Different users and uses dictate different levels of acceptability.

*__12.__ Page 14 refers to Figure 2. Again (a) and (b) are both not necessary – just have (b). Figure 3's caption should be revised to read "resulting from EAND X DFND" because this is not a flood hazard map but a map of that index. The topographic index defined by the authors contributes to*

*one kind of flooding but there are others that are equally as hazardous that are not well represented. British Columbia suffers from severe flash flooding that moves enormous amounts of debris yet there seem to be few hazards associated with this type of flooding that is mostly in mountanous regimes showing up in the map because of the way the authors have chosen their index. Can the authors comment on the universality of their choice in Canada? The authors clearly state early in their paper that extreme flooding in Canada is the result of many factors like ice jams, etc. This is very true and thus, the index defined by the authors cannot in fact be toted as a flood hazard by virtue of the fact that what leads to sudden high streamflow – the really danger - is not simply a flat area close to a stream bank. But if that's what the authors want to create, that's okay but then it requires a good discussion of why the approach is novel for defining a flood plain and what the benefits are (like computational ease), then they need to report the computational cost of creating these maps and report a quantitative comparison with things like the 1/100 year flood plain map in Calgary. Figure 5 refereed to on page 15 shows areas of overlap between the product and the flood plain map. This is again qualitative. A more quantitative comparison is required with even something simply like number of grid cells overlapped versus not overlapped to start with.*

**R12.** Figure 2 (a) and (b) help the readers see the difference that reclassification into 5 classes cause to the map. As discussed earlier, we disagree on the issue of Hazard definition. The issue of debris from the mountain can be just another index added to Hazard based on proximity to erodible mountains in the headwaters. However, it should be noted that the commonly used, and widely accepted, flood hazard maps based on flood frequency do not take into account factors, such as debris flow. The issue of ice jamming is true as a cause of flooding, but inundated areas first impacted are the lowlands and lands close to the stream, we cannot see how ice jams negates the universality of our proposed hazard index. Our approach simply prioritizes areas expected to be flooded first, then second, and so on.

*__13.__ This brings me to my next point. Large municipal urban centres already have information on high flood risk regions. What information does this product bring them that they don't already have at a better resolution? Risk of fire is largely a problem when it starts encroaching on an urban area and not generally at the same time as a flood risk so how can this low resolution product be helpful to Calgary?*

**R13.** Based on this revised manuscript with DEM resolution of 20 m, we believe that this product is no longer a low resolution product. We believe that we addressed this point earlier, and also in the manuscript. What an approach like this brings is different. This approach helps prioritize areas for detailed modeling, help development planning, and other studies, such as investigation of various population groups and their vulnerability to certain hazards, which is useful for resource allocation. Recognizing the areas exposed at high flood hazard is a urgent priority in many regions of the world, we believe that our approach is a significant step forward.

*__14.__ The discussion is lacking in many regards in this paper particularly where figures are produced. Page 16 for example refers to figure 6 but honestly, there is nothing really discussed or noted of significance here. Figure 7 is too coarse a resolution to be useful. Figure 8 is an "enlarged" version of an area for better visual interpretation but if they don't provide the exact area in space (not just with hatchmarks but perhaps with an areal photo showing the flood plain in the area) it is not a useful figure. This figure also has little discussion.*

__R14.__ We believe that enough discussion is provided for Figure 6 (New figure 5). Discussion on Figure 7 has also been expanded (page 19, line 23 – page 20 – line 4) in the revised manuscript. While providing maps for a large country like Canada, the details are lost in the main figure. We have enlarged a small area and presented it to improve the interpretation. Addition of aerial photos over risk map (Figure 8) results in too many layers that would not be easily interpreted. Hence we avoided including additional layers.

*__15.__ The authors don't provide a rigorous enough evaluation of their product at this stage. In Figure 10, the authors refer to reduced levels of social risk for commercial regions. Again I disagree with this but perhaps this is due to a lack of rigerous definitions on the part of the authors as to what is "social" – human residential impairment? The authors should revise all their captions to state what is truly shown. Also, there were numerous areal photos of flooded regions within Calgary during the 2013 floods. Why not use this valuable information to compare to their product? That would be a much better evaluation and would demonstrate the deficiencies and limitations of the product in an actual flood that was not 1 in 100 but with an extent that was outside the 1/100 year flood plain.*

__R15.__ The issue of population in residential and commercial/industrial areas was discussed earlier. We compared with 100-year flood modeling in Calgary because a georeferenced map was made available to us, and the hydraulic modeling is supposed to be the "accurate and scientific" way of mapping flood inundation. Therefore, enough information is available to perform a validation of our results. However, we also managed to obtain a high resolution aerial photo of the 2013 flood in the Qu'Appelle River Basin, which is one of the most challenging areas in Canada for hydrologic and hydraulic modeling – the Canadian Prairies. We compared our product with this photo and the results are very good, and this is now presented and discussed in the revised manuscript (Page 18, lines 13-21, Figure 5). We thank the reviewer for bringing this up because it gave us the opportunity to conduct another compelling validation of our approach and product.

*__16.__ Page 17: line 17, the authors refer to the "average" effect. Why would they be integrated in the first place? Why is "average" in quotes? My point is that this work is really a GIS exercise and the GIS community understands the issues and limitations with combining data of different resolutions, etc., yet I'm concerned with the lack of attention to terminology or basic GIS concepts used in the discussion. A more formal language is preferred along with greater detail on what was actually created and how.*

**R16.** Simply what we meant is that integrating two aspects in one can mask the individual effects. Sometimes integration is a must, and we did it, for example, with EAND and DFND, but in case of effect on population we wanted it to be explicit. As suggested, and as discussed earlier, additional details on the GIS-related analysis were provided in the revised manuscript with regard to handling maps with different resolutions and the accuracy of the metadata. We agree that the application of the work is basically a GIS exercise, which however makes use of an innovative idea and provides innovative information.

*17. I really do think products like these are good ideas but it's not just what is novel that must be shown but how it is useful and why it is needed. Unfortunately, I do not feel that the reader is given a full understanding of how this approach or product is useful. There is some attempt but more depth is needed. For example, on page 18, line 15, the authors state: "In other regions, and depending on the topography, the 100 year flood might cover two or three of the flood hazard classes." I don't mean to sound curt but so what? How is this useful to a planner that is required by most by-laws to deal with the 100 year flood or design with the 5, 10 or 30 year flood in Canada? Typographical errors: Line 13, Page 6 – insert "data" after "remotely sensed" Page 8 – insert "an" or "the" before "eight" Page 11 – line 9 replace "from" with "for" Page 32: Spelling eerror in the caption of Figure 8 (should be severe not sever)*

**R17.** We believe that our approach is indeed presenting an original contribution, and we also believe that it is very useful. It allows the identification of critical areas, where subsequent detailed analyses should focus on. For example, local authorities may want to relate flows at different flood frequencies (e.g., 100 year) to water stage (can be done using rating curves available locally). The stage of different floods will indicate clearly which of our hazard classes will be inundated. This way local authorities can convert our map to flood frequencies. This clarification was added to the revised manuscript on Page 22, lines 6-10. Typographical errors were corrected.

**Reviewer 2**

The authors would like to thank Reviewer #2 for providing a review. We are providing here below our detailed response to each remark. Some of our remarks are copied from our response to the first Reviewer wherever the reviewer's comment is similar to one made by the first reviewer.

*1. I miss a clear statement of the research problem and what is novel with the purposed study. The structure of section one and two could be improved by avoiding jumping back and forth between topics.*

**R1.** The paragraph on Page 2, Lines 15-23 and Page 4, Lines 7-12 state clearly the problem and the aim of our work. We would like to emphasize that our approach here proposes for the first time the integration of detailed topographic information, in the form of distance and elevation from streams, with hydrologic and human settlements information to assess flood risk. We cannot see eye to eye with the reviewer the issue of jumping between topics, however, we reviewed this carefully, and we could not identify the problem.

*2. [Page 8-9] To create the EAND and DFND classes, a drainage network was created using ArcGIS hydrology tool on a coarse resolution DEM. This can produce many errors - why not use an already existing drainage network, or at least verify against one?*

**R2.** In the revised manuscript, we presented everything using the finest scale-resolution DEM available for Canada (20 m). Even the river network made available through Environment and Climate Change Canada (ECCC) is generated using DEMs of a coarser resolution. We used Google Earth to compare the river network we generated against actual rivers, and the comparison, which validates the use of the 20 m DEM, was already shown in our response to Reviewer 1.

*3. [Page 9, Lines 12-13] The classification process for the different maps produced is not clear. For example, the hazard class intervals were selected somewhat arbitrarily. I would like to see more thought behind this, e.g., do they represent floodplains, and why five classes?*

**R3.** The five hazard classes selected can represent different hazard levels across the country as the topography is different across the country. However, as we explained on Page 18, this can be adapted locally to different types of representation; e.g. flood frequency. As the reviewer pointed out, the intervals for DFND and EAND were decided taking into consideration that flooding extent in floodplains would be much larger than in hilly areas. In hilly areas, EAND governs the hazard mapping, thus reducing the extent of hazard. For the study over the entire country, the 5 classes considered were deemed adequate.

*4. [Page 12, Lines 6-8] The exposure map based on nightlight data indicate that 98% of Canada's area has absent or low human activity. This leads to the following question –is a national flood risk assessment useful?*

**R4.** Yes, very useful. Majority of the population is in southern areas, however, hundreds of thousands of Canadians are spread across the Canadian landscape. Some of the northern population groups can be even more socially vulnerable than others, and floods in their regions are critical. In addition, major infrastructures, including roads which are important element for mobilizing rescue efforts are spread across what seems to be areas with low nightlight luminosity, which is now presented by the modified flood hazard and exposure maps in the revised manuscript (Figures 3

and 7). This work aims to highlight these issues. Because of the large area of Canada (almost equivalent to the area of Europe), visually, it looks like most of the country is dark at night, but zooming in can reveal more details. The availability of our product in a digital form with 20 m resolution allows for investigating issues at finer scales.

*5.* *[Page 12, Table 2 and 3; Page 31, Figure 7; Page 32, Figure 8] The land-use classes and the nightlight classification used for the exposure map give northern communities very low or low exposure level by default, resulting in very low or low flood exposure, and very low flood risk in areas above $60^o$ N. Is this national flood risk map useful for residents above $60^o$ N? I am missing a discussion around how the classification process affects the end product.*

**R5.** The first part of the question was addressed by our response to the previous comment. The classification process and selection of number of classes are usually arbitrary and subject to the judgement of the analysts. However, it is more convenient to fix the number of classes of the various maps. Increasing the number of classes would not be of much help as decision makers would eventually prefer to lump a few intermediate classes for easier interpretation. Five land-use classes are sufficient as one can even associate an average dollar value to each class.

*6.* *[Page 14-15, Lines 14-21, 1-5] A coarser DEM is chosen for the study to keep computational costs low, but results show that a finer resolution DEM (20 m in this case) gives better results and a more reliable flood hazard assessment. Floods are usually analyzed and managed at the provincial level in Canada where local information is important, why is a national flood risk assessment needed?*

**R6.** The idea is a way to address flood risk at large scale, and we addressed the importance of this earlier. Even at the provincial level where one province in Canada (e.g., Quebec is more than twice the area of France) is too large for detailed flood mapping based on hydraulic modeling. Our approach is useful even at the provincial level, especially in light of the fact that we reproduced the maps using 20 m resolution.

*7.* *[Page 15, Lines 16-20] It is suggested that hazard levels can be reclassified locally to match floods with different return periods in areas where flood inundation using hydraulic modeling is available. But, how useful are local topography-based flood hazard maps where flood inundation maps based on hydraulic modeling already exist? Also, topography-based flood hazard maps does not account for backwater and other hydraulic effects on areas upstream of flood protection. One related question is also how useful flood hazard maps with different return periods are if many floods are caused by ice-jams [Page 7, Lines 7-8; Page 18, Lines 11-14]?*

**R7.** We meant that areas where hydraulic modeling was done can be used as key locations to identify the water stage that corresponds to certain flood frequencies, which can be also simply approximated using rating curves). When flood stages of different flood frequencies are estimated, they can replace our hazard classes. Perhaps our map can be also used for practical extrapolation over larger areas based on finding match between our map and modeling-based inundation maps at some key locations. The issue of backwater curve not captured by our approach is certainly acknowledged in our manuscript (Page 19, lines 10-14).

Ice jams do cause floods. However, this is not a universal phenomenon. At locations where information on ice-jams are available, floods can still be translated to flooding depths and the same map can be used to determine the associated hazard upstream of it, independent of return-periods.

***8.** [Page 16-17, Lines 23-24, 1-3] The authors bring up the issue with overglow effect when analyzing nightlight data. Have potential overglow effects been analyzed for the 2013 nightlight data used in this study, e.g., in comparison with previous years?*

**R8.** Overglow effect is inherent with nightlight images for all years. We did not carry out any comparison study on overglow variations in nightlight images as the decision was to use the latest nightlight imagery for the study. The classification of DN into different classes alleviates the overflow effect to some extent.

***9.** [Page 17, Lines 10-19] There is a discussion that population data should be used together with nightlight data to separate social and economic impact, as airports and industrial areas show high luminous values but low population density. I will argue that although these built-up areas have low population density, they have high social impact, e.g., airports.*

**R9.** We agree with the reviewer. The purpose here was to show that using only census data might not be enough to determine social impacts as zero population according to census do not mean no human presence.  There is still capital investment and human lives, which are disturbed at different levels when homes, workplace, or transportation are impacted. A sentence about this was added in the revised manuscript (Page 15, lines 17-18).

***10.** [Page 19, Lines 12-16] There are many uncertainty aspects with the classes identified and some of the methods used – is the final product really useful and practical [Page 20, Lines 6-7] - also when considering the shortcomings the authors have presented?*

**R10.** The classes identified and methods used do have a degree of uncertainty with them and we have identified and provided discussion on them in the manuscript. The final product is still useful and practical as it is easy to obtain these maps for any part of the country. The shortcomings do not affect the methodology as much as it affects the end product. With the provided approach, the product can always be subject to improvements when finer/accurate data become available.

However, it should be noted that in this revised manuscript, the product itself is improved as it was redeveloped using much finer-resolution DEM, and both quantitative and qualitative validation are provided in the revised manuscript.

*__11.__ The article has 10 figures, are all of them needed? For example, Figure 1 – a and b should be combined if to be included at all. Also, is both a and b in Figure 2 needed, they show the same information. Figure 5 – exclude enlarged figures, and visually improve the main figure.*

__R11.__ Yes, we agree regarding combining Figure 1 a and b in one piece, and we did so in the revised manuscript. Figure 2 is important to show, at least visually, the effect of classification of nightlights. As for figure 5 (New figure 4), the enlarged portions are shown to discuss visual comparison and is now further supported by quantitative comparison.

*__12.__ Minor issues: [Page 1, Line 13] The authors state that the study uses datasets at reasonably fine resolutions to create flood risk maps – what is considered reasonable?*

__R12.__ This statement was modified to "The study focuses on using global and national datasets available with various resolutions to create flood risk maps."

*__13.__ [Page 9, Line 4] What do you mean by horizontal distance?*

__R13.__ We are referring to a buffer like distance while describing DFND. However, in GIS, the term "buffer" is usually applied to concentric distances to a feature (line, point or polygon) in vector format. For the present study, the stream network was retained in raster format to maintain consistency in all subsequent calculations. Horizontal distance refers to the Euclidean distance between the drainage cells and adjoining cells that are estimated using the "Euclidean distance" tool in ArcGIS, followed by reclassification using the limits mentioned in Table 1. Hence, the word "buffer" was avoided and "horizontal distance" was used instead.

*__14.__ [Page 9, Line7] EAND instead of EFND*

__R14.__ Thanks, we corrected it.

*__15.__ [Page 11, Lines 19-22] It is stated that the average values of all nightlight satellites were used in this study, but there is only one available for 2013.*

__R15.__ As the reviewer pointed out, data for 2013 is only from a single satellite. The sentence referring to this was modified in the revised manuscript (Page 13, line 21). The availability of data from more than one satellite is true for some years for which data are available and we were referring to that.

*__16.__ [Page 17, Line 17] What is the "average" effect?*

__R16.__ Simply what we meant is that integrating two aspects in one can mask the individual effects.

*__17.__ [Page 21, Line 31] De Moel should be de Moel.*

__R17.__ It is now corrected in the revised manuscript.

*__18.__ [Page 23, Line 25] The reference Schanze is not found in the text.*

__R18__. It was removed from the reference list in the revised manuscript.

[revised manuscript text omitted]

---

## Author Response (AR2)

Response letter to Reviewer 2 of "Topography- and nightlight-based national flood risk assessment in Canada"

The authors would like to thank the anonymous reviewers for providing a second review of the manuscript.

***1.*** *Thank you for such a careful revision. I only have a few minor comments:*
*Page 2, line 20: Was 2011 chosen as an example for being a particular bad year?*

**R1.** Yes, globally, and that is why we wrote "for example" on Line 20. We also mentioned the 2016 flood in Louisiana on Line 21.

***2.*** *Page 3, lines 7-8: I would suggest adding in the text that "flood risk as a product of hazard, exposure, and vulnerability" is brought up in detail in section 2.*

**R2.** Corrected as suggested

***3.*** *Page 3, lines 12-13: This sentence can perhaps move up before lines 8-11, or edit lines 12-13 to emphasize that this statement belongs to lines 8-11.*

**R3.** Thank you, the sentence was moved up as suggested.

***4.*** *Page 9, lines 1-4: Perhaps mention that both resolutions were assessed in the study? I don't find this information until the results section.*

**R4.** Added to Lines 7-8, Page 9.

***5.*** *Page 10, lines 10-13: Consider rephrasing this long sentence, e.g., can "by the city of Calgary" be included in the reference (Government of Alberta, 2013)?*

**R5.** We shortened the sentence a bit by removing the last part of it as it was not really necessary.

**LIST OF CHANGES MADE IN THE MANUSCRIPT:**

Only the requested editorial changes were made on Pages 4, 9, and 11 of this document.

[revised manuscript text omitted]